

# Determination of muscle strength and function in plesiosaur limbs: finite element structural analyses of *Cryptoclidus eurymerus* humerus and femur

Anna Krahl[1,2,3], Andreas Lipphaus[2], P. Martin Sander[1] and Ulrich Witzel[2]

[1] Institute of Geoscience, Section Paleontology, Rheinische Friedrich-Wilhelms Universität Bonn, Bonn, Germany
[2] Biomechanics Research Group, Chair of Product Development, Faculty of Mechanical Engineering, Ruhr-Universität Bochum, Bochum, Germany
[3] Paleontological Collection Fachbereich Geowissenschaften, Eberhard-Karls-Universität Tübingen, Tübingen, Germany

Corresponding author
Anna Krahl,
annakrahl1@googlemail.com

## ABSTRACT

**Background:** The Plesiosauria (Sauropterygia) are secondary marine diapsids. They are the only tetrapods to have evolved hydrofoil fore- and hindflippers. Once this specialization of locomotion had evolved, it remained essentially unchanged for 135 Ma. It is still controversial whether plesiosaurs flew underwater, rowed, or used a mixture of the two modes of locomotion. The long bones of Tetrapoda are functionally loaded by torsion, bending, compression, and tension during locomotion. Superposition of load cases shows that the bones are loaded mainly by compressive stresses. Therefore, it is possible to use finite element structure analysis (FESA) as a test environment for loading hypotheses. These include muscle reconstructions and muscle lines of action (LOA) when the goal is to obtain a homogeneous compressive stress distribution and to minimize bending in the model. Myological reconstruction revealed a muscle-powered flipper twisting mechanism. The flippers of plesiosaurs were twisted along the flipper length axis by extensors and flexors that originated from the humerus and femur as well as further distal locations.
**Methods:** To investigate locomotion in plesiosaurs, the humerus and femur of a mounted skeleton of *Cryptoclidus eurymerus* (Middle Jurassic Oxford Clay Formation from Britain) were analyzed using FE methods based on the concept of optimization of loading by compression. After limb muscle reconstructions including the flipper twisting muscles, LOA were derived for all humerus and femur muscles of *Cryptoclidus* by stretching cords along casts of the fore- and hindflippers of the mounted skeleton. LOA and muscle attachments were added to meshed volumetric models of the humerus and femur derived from micro-CT scans. Muscle forces were approximated by stochastic iteration and the compressive stress distribution for the two load cases, "downstroke" and "upstroke", for each bone were calculated by aiming at a homogeneous compressive stress distribution.
**Results:** Humeral and femoral depressors and retractors, which drive underwater flight rather than rowing, were found to exert higher muscle forces than the elevators and protractors. Furthermore, extensors and flexors exert high muscle forces compared to Cheloniidae. This confirms a convergently evolved myological

![PeerJ]

mechanism of flipper twisting in plesiosaurs and complements hydrodynamic studies that showed flipper twisting is critical for efficient plesiosaur underwater flight.

## INTRODUCTION

Plesiosauria are secondarily aquatic tetrapods that lived in the sea from the Late Triassic (*Wintrich et al., 2017*) until the Cretaceous–Paleogene (K–Pg) mass extinction (*Bardet, 1994*; *Motani, 2009*; *Vincent et al., 2011*; *Vincent et al., 2013*; *Bardet et al., 2014*). Plesiosauria are the most evolved group of Sauropterygia. They first appear in the fossil record in the late Early Triassic, but their origin and relationships remain unclear. Sauropterygia may have evolved from basal archosauromorphs (*Merck, 1997*) or lepidosauromorphs (*Rieppel & Reisz, 1999*), or may be a sister taxon to archosauromorphs and lepidosauromorphs (*Neenan, Klein & Scheyer, 2013*). The most striking and unique key innovation of plesiosaurs is that by the Late Triassic they had evolved four similar-looking wing-like flippers (*Wintrich et al., 2017*), which must have been their only propulsive organs, unlike all non-plesiosaurian sauropterygians, which were largely axial swimmers.

In plesiosaurs, all four pectoral flippers are dorsoventrally flattened, have a greatly shortened zeugopodium (<one-third of autopodium length) (*Wintrich et al., 2017*), and form a hydrofoil profile (*Robinson, 1975*, *1977*). The pectoral flippers may have had an asymmetrical profile (*Robinson, 1975*; *Caldwell, 1997*), like the pectoral flippers of present-day underwater flying penguins (Spheniscidae) and sea turtles (Chelonioidea (*i.e.*, Cheloniidae + Dermochelyidae)). Unlike plesiosaurs, only the foreflippers of penguins, sea turtles, whales and dolphins (Cetacea), and sea lions (Otariinae) are hydrofoil-like flippers (*Walker, 1973*; *Schreiweis, 1982*; *Feldkamp, 1987*; *Fish & Battle, 1995*; *Fish, 2004*; *Cooper et al., 2007*; *Elliott et al., 2013*). The hindflippers of sea turtles, penguins, and sea lions are paddles (*Shufeldt, 1901*; *Walker, 1971b*; *Davenport, Munks & Oxford, 1984*) which are used for steering and locomotion on land and in water (*Walker, 1971b*; *Pinshow, Fedak & Schmidt-Nielsen, 1977*; *Clark & Bemis, 1979*; *Davenport, Munks & Oxford, 1984*; *Feldkamp, 1987*; *Wyneken, 1997*). The hind limbs of cetaceans were completely reduced during their evolutionary history.

### Osteology of the appendicular skeleton of plesiosaurs

The scapula and coracoid of the pectoral girdle, and the pubis and ischium of the pelvic girdle of plesiosaurs, consist of large flat bony plates lying ventrally. All except the scapula meet the element on the other side in a medial symphysis at the midline of the body in a slightly v-shaped configuration. In some plesiosaurs, the scapulae also meet in a medial symphysis. The coracoid and pubis are greatly expanded. The dorsal extension of the scapula and the dorsally directed ilium are greatly reduced (*Andrews, 1910*). Gastralia lie

between the pectoral and pelvic girdles and stiffen the trunk region (*Sues, 1987*; *Taylor, 1989*). Humeri and femora of some plesiosaurs such as *Cryptoclidus eurymerus*, a Middle Jurassic cryptoclidid known from many specimens from the lower Oxford Clay Formation of England (*Andrews, 1910*), have a rounded proximal end and an oval midshaft cross-section. The epicondyles are expanded and hammer-shaped. The mounted skeleton of *Cryptoclidus eurymerus* in the Goldfuß Museum, University of Bonn (IGPB R 324) shows these features well. The humeral and femoral heads of plesiosaurs were probably covered by a thick and vascularized cartilaginous cap (*Krahl, 2021*), as has been reported for *Dermochelys coriacea* (*Rhodin, Ogden & Conlogue, 1981*; *Snover & Rhodin, 2008*).

The distal end of the humerus of *Cryptoclidus* is greater than that of the femur and rather atypical for most plesiosaurs (see, *e.g.*, *Großmann, 2006*; *Sachs, Hornung & Kear, 2016*). Radius/ulna and tibia/fibula are greatly shortened and have a discoid appearance (*Andrews, 1910*). Metacarpal V and metatarsal V are integrated proximally into the rows of carpals and tarsals (*Robinson, 1975*). Fore- and hindflippers exhibit hyperphalangy (*Andrews, 1910*).

## Flipper muscle reconstructions

Plesiosaur muscles have been previously reconstructed by several researchers, including *Watson (1924)*, *Tarlo (1958)*, *Robinson (1975)*, *Lingham-Soliar (2000)*, *Carpenter et al. (2010)*, *Araújo & Correia (2015)*, and *Krahl & Witzel (2021)*. *Carpenter et al. (2010)*, *Araújo & Correia (2015)*, and *Krahl & Witzel (2021)* relied on the extant phylogenetic bracket (EPB), a method developed in the 1990s to make reliable inferences about the soft tissue anatomy of fossils (*Bryant & Russel, 1992*; *Wittmer, 1995*). Older studies (*Watson, 1924*; *Tarlo, 1958*; *Robinson, 1975*; *Lingham-Soliar, 2000*) did not use EPB and did not clearly indicate which living taxa they relied on for their muscle reconstructions. Muscles originating at the pectoral girdle and attaching to (or spanning) the humerus were reconstructed by *Watson (1924)*, *Tarlo (1958)*, *Robinson (1975)*, *Lingham-Soliar (2000)*, *Carpenter et al. (2010)*, *Araújo & Correia (2015)*, and by *Krahl & Witzel (2021)*. Locomotory muscles originating from the pelvic girdle, traversing the femur, and attaching to the femur were partially reconstructed by *Robinson (1975)*, *Lingham-Soliar (2000)*, and *Carpenter et al. (2010)*, and completely reconstructed by *Krahl & Witzel (2021)*. Muscles originating distally from the humerus and femur were partially reconstructed by *Robinson (1975)* and completely by *Krahl & Witzel (2021)*. *Robinson (1975)* appears to have reconstructed the ventral side of the foreflipper and the dorsal side of the hindflipper, although this is not clearly stated (*Robinson, 1975*; *Krahl & Witzel, 2021*). The superficial conclusion would be that the dorsal and ventral fore- and hindflipper sides might look the same in plesiosaurs, which is not supported by the EPB because the dorsal and ventral fore and hind limb musculature of living Sauropsida are not congruent (*Walker, 1973*; *Meers, 2003*; *Russell & Bauer, 2008*; *Suzuki et al., 2011*). The distal fore- and hindflipper musculature of plesiosaurs reconstructed by *Robinson (1975)* looks very similar to the foreflipper musculature of cetaceans in its extreme reduction (*Cooper et al., 2007*). This whale-like condition seems unlikely for plesiosaurs because the foreflippers of whales are merely control surfaces and not lift-producing propulsory organs (*Fish, 2002*; *Woodward,*

*Winn & Fish, 2006*), whereas the main propulsive organ of whales is a large muscular swimming tail with a fluke (*Fish, 1996*; *Woodward, Winn & Fish, 2006*). In contrast, plesiosaurs swam actively with their fore- and hindflippers (*Krahl & Witzel, 2021*). *Krahl & Witzel (2021)* were the first to reconstruct the entire locomotor musculature of the fore- and hindflippers of a plesiosaur. They reconstructed a complex arrangement of muscles for plesiosaur fore- and hindflippers that would allow the plesiosaur to twist its two pairs of flippers along the flipper length axis and perhaps even actively control the flipper profile, as suggested by hydrodynamic calculations of plesiosaurs by *Witzel, Krahl & Sander (2015)* and *Witzel (2020)*.

## Hypotheses on the locomotion of plesiosaurs

The locomotion style of plesiosaurs has been the subject of an ongoing debate for over a century (*Williston, 1914*; *Watson, 1924*; *Tarlo, 1958*; *Robinson, 1975*, *1977*; *Feldkamp, 1987*; *Lingham-Soliar, 2000*; *Carpenter et al., 2010*; *Araújo et al., 2015*; *Araújo & Correia, 2015*; *Liu et al., 2015*; *Krahl & Witzel, 2021*). It has been suggested that plesiosaurs rowed like ducks or otters (*Williston, 1914*; *Watson, 1924*; *Tarlo, 1958*; *Araújo et al., 2015*; *Araújo & Correia, 2015*), fly underwater like sea turtles and penguins (*Robinson, 1975*, *1977*; *Lingham-Soliar, 2000*; *Carpenter et al., 2010*; *Liu et al., 2015*; *Muscutt et al., 2017*; *Krahl & Witzel, 2021*), or use rowing-flight like sea lions (*Feldkamp, 1987*; *Liu et al., 2015*). The main difference between the different modes of locomotion stems from the underlying hydrodynamics: rowing uses water resistance to push the body forward, whereas underwater flight generates lift and thrust from oncoming water currents moving around a cambered flipper profile (*e.g.*, *Baudinette & Gill, 1985*; *Fish, 1996*; *Walker & Westneat, 2000*). Sea lion rowing-flight relies on both hydrodynamic mechanisms, drag and lift, at different stages of the limb cycle (*Feldkamp, 1987*). It has also been proposed that the anterior pair of flippers uses a different mode of locomotion than the posterior pair of flippers (*Tarlo, 1958*; *Lingham-Soliar, 2000*; *Liu et al., 2015*).

A flipper used in rowing is usually moved in an anteroposterior direction, with little dorsoventral movement (*Pace, Blob & Westneat, 2001*; *Rivera, Rivera & Blob, 2011*; *Rivera, Rivera & Blob, 2013*). The anteroposterior extension and dorsoventral reduction of the bony elements of the pectoral and pelvic girdles of plesiosaurs, and the accompanying reduction or hypertrophy of the locomotor musculature, have been interpreted as favoring the protraction and retraction of the pectoral flipper, *i.e.*, a rowing motion (*Watson, 1924*; *Tarlo, 1958*; *Godfrey, 1984*).

In contrast, the pectoral flipper is flapped mainly dorsoventrally during underwater flight, with a minor anteroposterior component. The downstroke of the flippers of Cheloniidae and Spheniscidae is characterized by strong depression and slight retraction of the flipper. The upstroke is accomplished by elevation and protraction of the humerus. The flipper tip describes an oblique "O" in the anterodorsal-posteroventral direction (*Clark & Bemis, 1979*; *Davenport, Munks & Oxford, 1984*; *Rivera, Wyneken & Blob, 2011*; *Rivera, Rivera & Blob, 2013*). As *Robinson (1975)* noted, the hydrofoil-shaped flippers of plesiosaurs, tapering toward the tip and superficially comparable to those of penguins and sea turtles, suggest that they were used for underwater flight rather than

rowing. In addition, the shoulder and hip joints of plesiosaurs restrict movement in the anteroposterior direction more than in the dorsoventral direction (*Krahl, 2021*).

In rowing-flight, the downstroke is lift-generating and similar to the downstroke of underwater fliers. At the point of maximum ventral flipper excursion, the flipper suddenly functions like a paddle and pushes against the water as it is retracted and elevated. During the recovery stroke, the flipper is protracted and elevated with little to no contribution to propulsion (*Feldkamp, 1987*). *Godfrey (1984)* noted that in recent underwater fliers, the shoulder girdles are characterized by a strong bony support that extends in a dorsoventral direction, which is not the case in plesiosaurs. Therefore, he concluded that sea lions and plesiosaurs share more similarities than recent "true" underwater fliers (*Godfrey, 1984*).

In addition to locomotion style, there is still debate about how the four flippers moved in relation to each other, *i.e.*, fore- and hindflippers synchronous, asynchronous, or out of phase. This is the so-called "four-wing problem", which is about how plesiosaurs avoided placing their hindflippers in the vortices shed by their foreflippers, which would result in a significant decrease in hindflipper performance (*Frey & Riess, 1982*; *Tarsitano & Riess, 1982*; *Lingham-Soliar, 2000*; *Carpenter et al., 2010*; *Muscutt et al., 2017*). Experiments in a water tank showed that plesiosaur underwater flight was most efficient when the fore- and hindflippers were slightly out of phase, *i.e.*, the foreflipper was slightly in front of the hindflipper (*Muscutt et al., 2017*).

## Basics of muscle physiology

During a limb cycle, muscles can follow either a shortening-stretch cycle or a stretch-shortening cycle. In the first case, the muscle is shortened while its force output increases. Then, the muscle is stretched and the force output decreases. In the latter, the muscle is initially stretched while its force output increases. As the muscle contracts, its force output decreases (*Rassier, MacIntosh & Herzog, 1999*). The force production of the muscle depends on the architecture of the muscle. Muscle architecture includes the length of the muscle and tendon (if any), lines of action, muscle mass, specific density of the muscle, intrinsic muscle force, fascial length, and pennation angle (*Alexander & Vernon, 1975*; *Gans, 1982*; *Sacks & Roy, 1982*; *Powell et al., 1984*; *Narici, Landoni & Minetti, 1992*; *Anapol & Barry, 1996*; *Kummer, 2005*; *Azizi, Brainerd & Roberts, 2008*). Parallel-fibered muscles have, on average, longer fascicles, greater volume, and can contract faster (which depends on the composition of the fiber types) than pennate muscles. However, more muscle fibers may be juxtaposed in a pennate muscle than in a parallel-fibered muscle of the same size. Therefore, a pennate muscle can exert a greater force than a parallel-fibered muscle of the same size, even though the pennate muscle has shorter fibers. This is because maximum muscle force depends not only on fiber length, but also on its physiological cross-sectional area (PCSA), *i.e.*, the sum of fiber cross-sections (*Gans, 1982*; *Burkholder et al., 1994*; *Allen et al., 2010*; *Huq, Wall & Taylor, 2015*).

The muscle length of a parallel-fibered, fast-contracting muscle also varies considerably and comes at the expense of a high metabolic cost and relatively low force production. In contrast, highly pennate muscles exert relatively high forces at a much lower metabolic

cost, but contract much more slowly and have poor control of fiber contraction (*Biewener & Roberts, 2000*).

Pennate muscles can generate relatively high forces while their muscle lengths behave almost isometrically, *e.g.*, in turkey (musculus (m.) gastrocnemius) (*Roberts et al., 1997*) or wallaby (m. gastrocnemius, m. plantaris) (*Biewener, Konieczynski & Baudinette, 1998*). In contrast, the pennate m. pectoralis muscle, which has a very high output, shows a very large muscle length change of 30–40% in *Columba livia* (*Biewener, Corning & Tobalske, 1998*; *Biewener & Roberts, 2000*) or about 30% in *Anas platyrhynchos* (*Williamson, Dial & Biewener, 2001*). In general, the change in muscle length of striated vertebrate muscles can be up to +/– 25% (contraction and extension relative to resting length (= 0%)) before their ability to generate force decreases significantly (*Biewener & Roberts, 2000*).

## Finite element structural analysis

As measurements with strain gauges show, the long bones of Tetrapoda are functionally loaded by torsion, compression, and bending in an intermittent fashion. Most often they are loaded by bending in alternate directions or mainly by compression and subordinately by tension. High torsional loads act on the long bones of terrestrial tetrapods (*Biewener & Dial, 1995*; *Carrano, 1998*; *Blob & Biewener, 1999*; *Main & Biewener, 2004*, *2007*; *Butcher et al., 2008*; *Butcher & Blob, 2008*; *Sheffield et al., 2011*; *Young & Blob*, *2015*; *Young et al., 2017*).

To represent such changing loading conditions, a series of load cases can be analyzed using finite element structural analysis (FESA; *Witzel & Preuschoft, 2005*) and then be superimposed on each other (*Carter, Orr & Fyhrie, 1989*). Reducing the bending moment (*Klenner et al., 2015*; *Lutz et al., 2016*; *Milne, 2016*; *McCabe et al., 2017*; *Lipphaus & Witzel, 2018*) can result in lightweight biological structures (*Klenner et al., 2015*). While structures under bending are subjected to compression on one side and tension on the opposite side, homogeneous compression with low tensile stresses indicates low bending. A further concept relevant for our analysis is bending-minimized lightweight design (*Witzel et al., 2011*; *Bartz, Uttich & Bender, 2019*; *Gößling et al., 2014*). The importance of bending-minimized lightweight design is that it reduces the energy required for locomotion. The principle of bending-minimized lightweight design is that the shape of the bone is guided by the functional superposition of all mechanical loads experienced by a given bone during its ontogeny (*Wolff, 1893*; *Carter, Orr & Fyhrie, 1989*; *Witzel & Preuschoft, 2005*). A homogeneous compressive stress distribution optimally utilizes the material: *Bartz et al. (2018)* conducted a study on a beam with a fixed support loaded with a single force at the end of the beam and showed a significant weight reduction when topology optimization and tension cables are combined to achieve homogeneous compressive stresses. We apply this principle in our FESA of the plesiosaur humerus and femur.

FESA is used in various disciplines that include engineering and biomechanics (*Rayfield, 2007*). FESA enables the analysis of mechanical stresses and strains in engineering and biological structures in 2-D or 3-D (*Rayfield, 2007*; *Witzel et al., 2011*) and

can contribute to our understanding of the function of bony elements (*Witzel et al., 2011*). Compressive loads are applied to bone *via* tension chords. Tension chords are either muscles and tendons (active tension chords) or ligaments (passive tension chords), and they act in pairs of agonists and antagonists (*Witzel & Preuschoft, 2005*; *Sverdlova & Witzel, 2010*; *Curtis et al., 2011*; *Witzel et al., 2011*; *Klenner et al., 2015*; *Felsenthal & Zelzer, 2017*, *Krahl et al., 2019*). A movement is driven by the agonist while the antagonist opposes it to perform a controlled movement (*Sverdlova & Witzel, 2010*). When the same movement is reversed, the agonist becomes the antagonist and vice versa. Thus, agonists and antagonists constantly load a bony structure by compressive stress, although the agonist exerts a proportionally higher force than the antagonist. In the past, FESA has been successfully used in functional morphology to predict tension chords in living animals, such as a collagenous and elastic tissue extending longitudinally below the septum nasi and associated with the m. pterygoideus anterior in crocodylians, which was later confirmed in dissections (*Klenner et al., 2015*).

The objective of this study was to test muscle reconstructions of plesiosaurs using FESA. Muscle reconstructions obtained with the EPB, *i.e.*, based on comparative anatomical studies, and mechanically imposed demands on muscles inform each other. We investigated how muscle physiological details, such as muscle length changes, can contribute to muscle reconstructions of fossils and whether FESA of the plesiosaur humerus and femur can provide information about plesiosaur locomotion. The results complement previous hydrodynamic and morphological studies suggesting that plesiosaurs must have used flipper twisting for efficient underwater flight (*Witzel, Krahl & Sander, 2015*; *Witzel, 2020*; *Krahl, 2021*), which is a common phenomenon in aquatic vertebrates (*e.g.*, *Walker & Westneat, 2000*; *Walker & Westneat, 2002*) and confirm the myological mechanism of flipper twisting reconstructed by *Krahl & Witzel (2021)*.

## MATERIALS AND METHODS

### Analog model of the lines of action of the flipper muscles of *Cryptoclidus*

LOA represent the direct connection in a straight line between the origin and attachment of a muscle (*Krahl et al., 2019*). However, it should be noted that this is a simplification made for FE/multibody dynamics modeling of often more complex muscle pathways that may, for example, wrap around bone during part of a movement cycle (*Krahl et al., 2019*; *Snively et al., 2013*). LOA were obtained experimentally in an analog model (Fig. 1) using resin casts of the pectoral (Figs. 2–4) and pelvic girdles and limbs (Figs. 5–7) of the mounted skeleton of *Cryptoclidus eurymerus* (IGPB R 324) exhibited at the Goldfuß Museum, Section of Paleontology, Institute of Geosciences, University of Bonn (IGPB), on which the muscle reconstructions of *Krahl & Witzel (2021)* are based.

To build the analog model, the casts of the pectoral and pelvic girdles were mounted on a wooden frame built from slats mostly 2.5 by 5 cm in cross section and fixed with screws. Slats were also used to model the vertebral column. The vertebral column needed to be part of the model because it serves as the origin of some locomotor muscles.

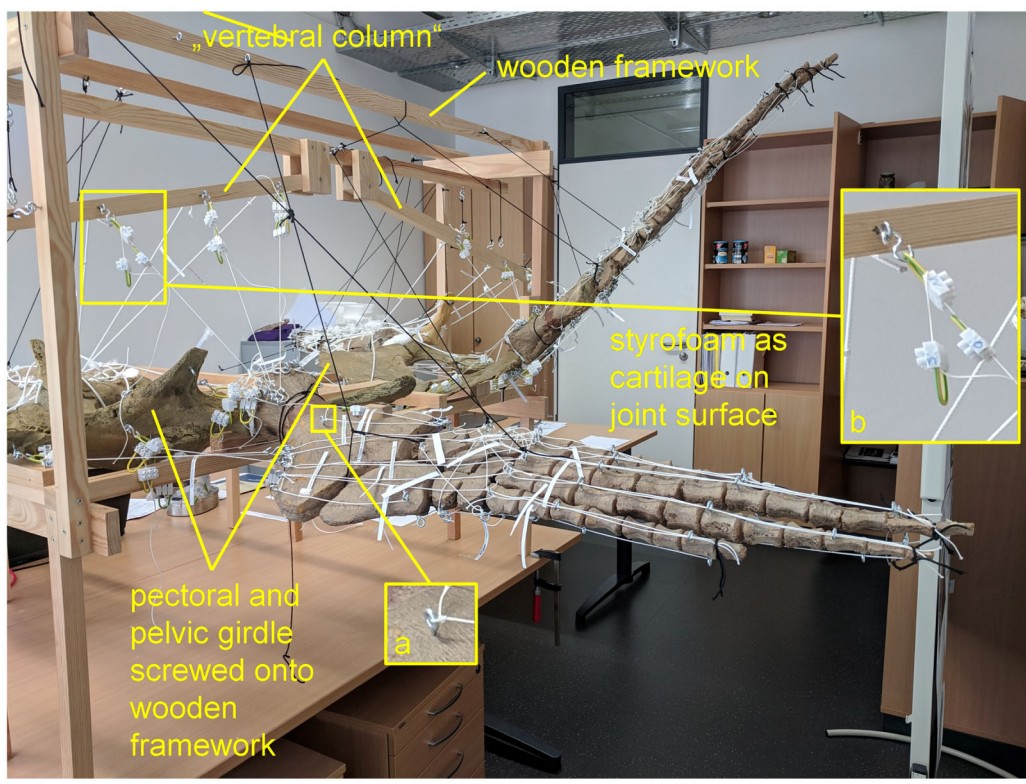

**Figure 1 Analog model demonstrating LOA of *Cryptoclidus eurymerus* (IGPB R 324) fore- and hindflipper.** Pectoral and pelvic girdle were fixed on a wooden frame. Thick styrofoam was placed into the glenoid and acetabulum joint cavity. Black threads helped to fix the flippers in their respective position. White threads represent LOA: a) screw eye pins were screwed into muscle attachment surfaces. b) Three electrical terminal strips were attached to one end. With hooks attached to each end of the thread, LOA were hung into the screw eye pins.

The anatomical positioning of the bone casts and the vertebral column was based on the mount. The missing cartilage cover in the shoulder and hip joints was replaced with thick styrofoam pads. The goal was to achieve congruency between the oval glenoid and acetabulum and the rounder humeral and femoral head. This required a styrofoam thickness of up to about ten millimeters. This assumption is based on the observation of specialized vascularized cartilage in *Dermochelys coriacea* and its associated osteological correlates (a highly grooved and pierced articular surface of the humeral head) (*Rhodin, Ogden & Conlogue, 1981*; *Snover & Rhodin, 2008*) and the presence of comparable osteological correlates in *Cryptoclidus eurymerus* humeri and femora, as noted by *Krahl (2021)*. Non-elastic white nylon cords were used to physically model the muscle lines of action, hereafter termed muscle cords. The fore- and hindflippers were hung into the frame and attached to the pectoral and pelvic girdle in the respective joints with help of the muscle cords. Screw eye pins were screwed into the cast at the muscle origin and insertion points, and the muscle cords were tied to these. When further mechanical support was necessary, the flippers were fixed in the selected flipper positions by additional nylon cords (Fig. 1).

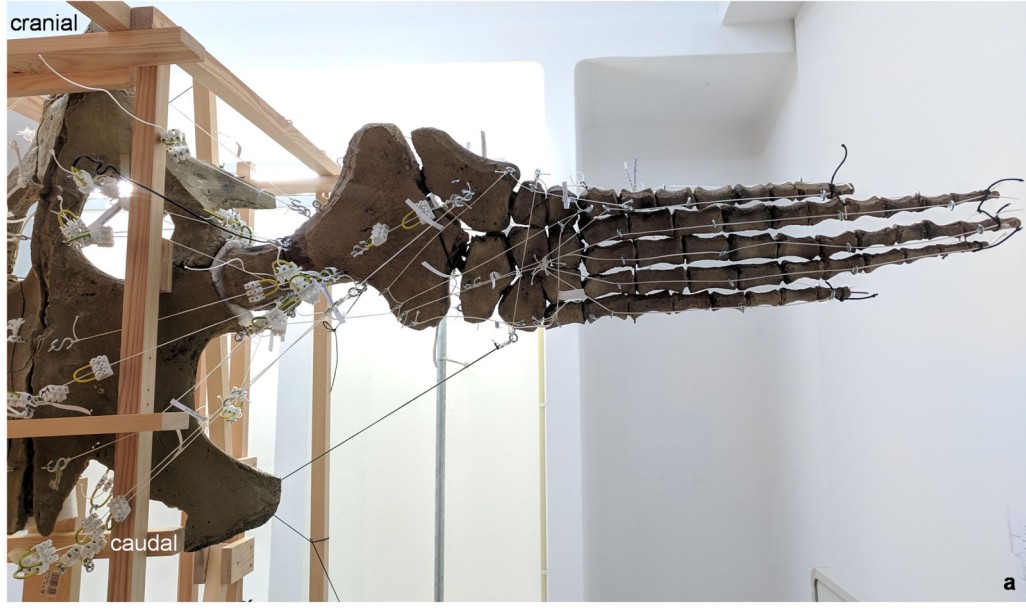

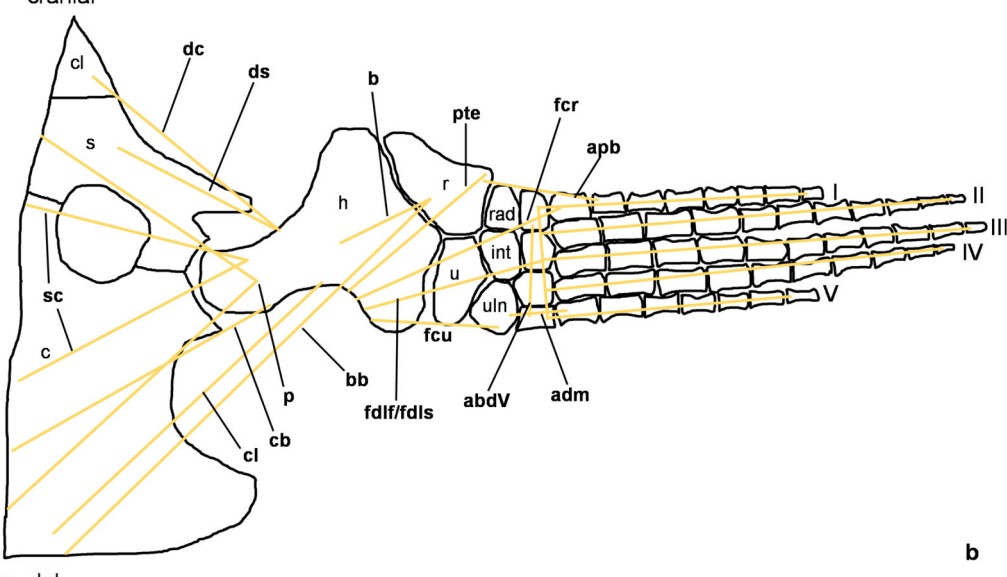

**Figure 2 Analog model of the myology of *Cryptoclidus eurymerus* (mounted skeleton IGPB R 324), shoulder girdle and foreflipper in ventral view.** (A) Mounted bone casts with white chords representing the lines of action of the foreflipper muscles. (B) Tracing of the shoulder girdle and foreflipper with muscle lines of action. Abbreviations of muscles: abdV, musculus abductor digiti V; adm, musculus adductor digiti minimi; apb, musculus abductor pollicis brevis; b, musculus brachialis; bb, musculus biceps brachii; cb, musculus coracobrachialis brevis; cl, musculus coracobrachialis longus; dc, musculus deltoideus clavicularis; ds, musculus deltoideus scapularis; fcr, musculus flexor carpi radialis; fcu, musculus flexor carpi ulnaris; fdlf/fdls, musculus flexor digitorum longus (foreflipper)/musculi flexores digitorum superficialis; p, musculus pectoralis; pte, musculus pronator teres; sc, musculus supracoracoideus. Abbreviations of bones: c, coracoid; cl, clavicle; h, humerus; int, intermedium; r, radius; rad, radiale; s, scapula; u, ulna; uln, ulnare; I, digit one; II, digit two; III, digit three; IV, digit four; V, digit five.               

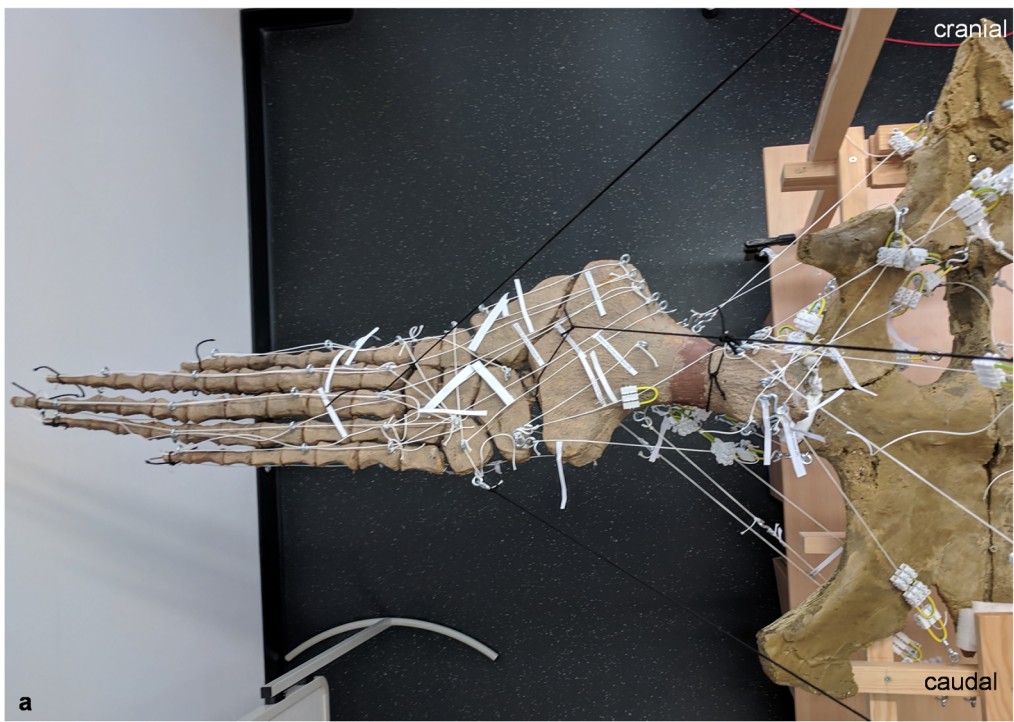

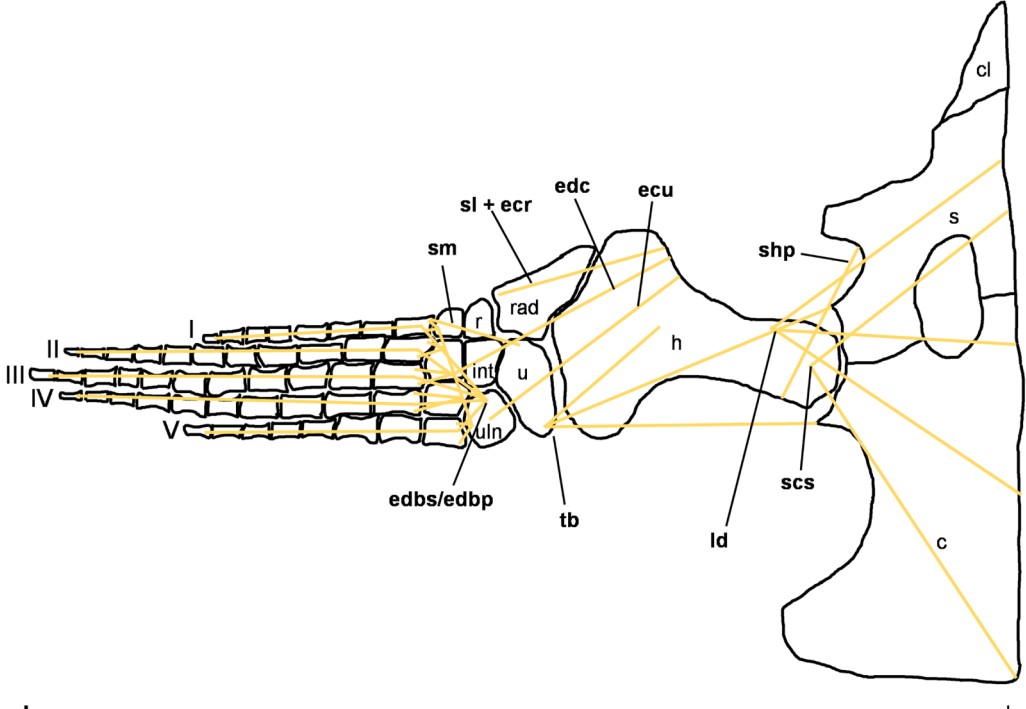

**Figure 3 Analog model of the myology of *Cryptoclidus eurymerus* (mounted skeleton IGPB R 324), shoulder girdle and foreflipper in dorsal view.** (A) Mounted bone casts with white chords representing the lines of action of the foreflipper muscles. (B) Tracing of the shoulder girdle and foreflipper with muscle lines of action. Abbreviations of muscles: ecu, musculus extensor carpi ulnaris; edbp/edbs, musculi extensores digitores breves profundi/musculi extensores digitores breves superficialis; edc,

**Figure 3** (continued)
musculus extensor digitorum communis; ld, musculus latissimus dorsi; scs, musculus sub-coracoscapularis; shp, musculus scapulohumeralis posterior; sl and ecr, musculus supinator longus and musculus extensor carpi radialis; sm, musculus supinator manus; tb, musculus triceps brachii. Abbreviations of bones: c, coracoid; cl, clavicle; h, humerus; int, intermedium; r, radius; rad, radiale; s, scapula; u, ulna; uln, ulnare; I, digit one; II, digit two; III, digit three; IV, digit four; V, digit five.

The insertions of all fore- and hindflipper locomotor muscles spanning either the glenoid or acetabulum (12 on the foreflipper, 14 on the hindflipper) were taken from *Krahl & Witzel (2021)* (Figs. 2, 3, 5, 6). Each muscle cord was threaded through the holes of three electrical terminal strips in sequence, later to be used in determining muscle length changes (Table 1). Hooks were then attached to both ends of the cords, which were then hooked into the screw eyes representing the muscle attachments (Fig. 1). If a muscle attachment area is small, the screw eye pin was screwed in approximately in the middle. In contrast, muscles with large attachment areas were divided into multiple subsections to better capture the different fiber attachment angles and directions and to avoid stress singularities that create artificially high stress and strain peaks at force transmission points (*Kim, Sankar & Kumar, 2018*). Accordingly, the total muscle force of a muscle was also subdivided, and portions of the total muscle force were assigned to the subsections. In general, the most anterior and posterior points in the body midline were chosen for screw eye pin placement (*e.g.*, m. subcoracoscapularis, m. coracobrachialis brevis, m. pectoralis). In the case of the m. latissimus dorsi, for example, a position between the most cranial and the most caudal origin was chosen. This was done to represent the parts that are best supported by the EPB (cranial and middle parts), but also to cover the less well supported part (caudal part) (*Krahl & Witzel, 2021*).

These muscle subsections do not necessarily represent actual partitions of the reconstructed muscles, although some muscles were likely compartmentalized, *e.g.*, m. pectoralis, m. latissimus dorsi, m. puboischiofemoralis internus, and m. puboischiofemoralis externus. Muscles with two or more heads (*i.e.*, m. deltoideus scapularis, m. deltoideus clavicularis, m. coracobrachialis brevis, m. coracobrachialis longus, m. triceps brachii, m. caudofemoralis brevis (ilium and vertebral column), m. flexor tibialis internus, m. flexor tibialis externus, m. puboischiofemoralis internus, and m. puboischiofemoralis externus) were implemented as two (or more) cords. The resulting vectors of the different subsections of these muscles, which have a large area of origin, were included in the FE models. Agonistic and antagonistic muscles attaching to, originating from, or spanning the humerus (Table 2) and femur (Table 3) were inferred from the mount. Images for documentation were taken from cranial/anterior (Figs. 4A–4F), caudal/posterior (Figs. 7A–7F), ventral (Figs. 2A and 5A), and dorsal (Figs. 3A and 6A). Figures 2, 3, 5, and 6 show some muscles that are not relatively close to the humerus, femur, or pectoral flipper. This is probably not an anatomically correct muscle course. Muscles are normally held closer to the body, such as by ligament slings. We did not add loop-like structures to measure muscle length changes in the analog model because it proved impractical. The implementation of the muscles in the FESAs were based on sketches in which we

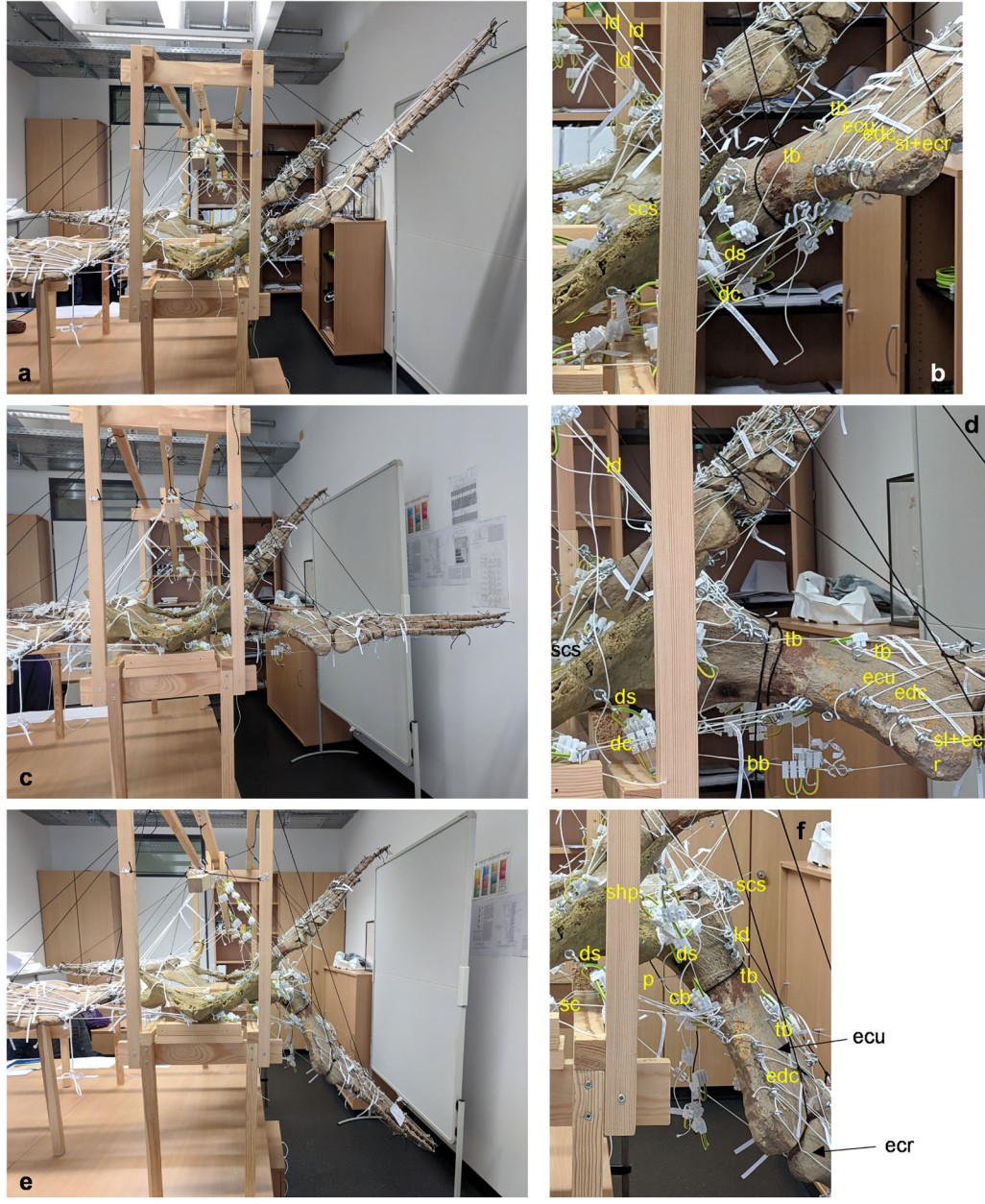

**Figure 4 Analog model of the muscle lines of action of the foreflipper of *Cryptoclidus eurymerus* (mounted skeleton IGPB R 324) in anterior and ventral views.** The images in the left column (A, C, E) show an overview of the flipper muscles. The images in the right column focus on the humeral muscles. (A) and (B) foreflipper during maximal dorsal excursion in anterior view. (C) and (D) foreflipper in the neutral position in the anterior view. (E) and (F) foreflipper during maximal ventral excursion in the anterior view. Abbreviations: bb, musculus biceps brachii; cb, musculus coracobrachialis brevis; dc, musculus deltoideus clavicularis; ds, musculus deltoideus scapularis; ecu, musculus extensor carpi ulnaris; edc, musculus extensor digitorum communis; ld, musculus latissimus dorsi; p, musculus pectoralis; sc, musculus supracoracoideus; scs, musculus subcoracoscapularis; shp, musculus scapulo-humeralis posterior; sl and ecr, musculus supinator longus and extensor carpi radialis; tb, musculus triceps brachii.

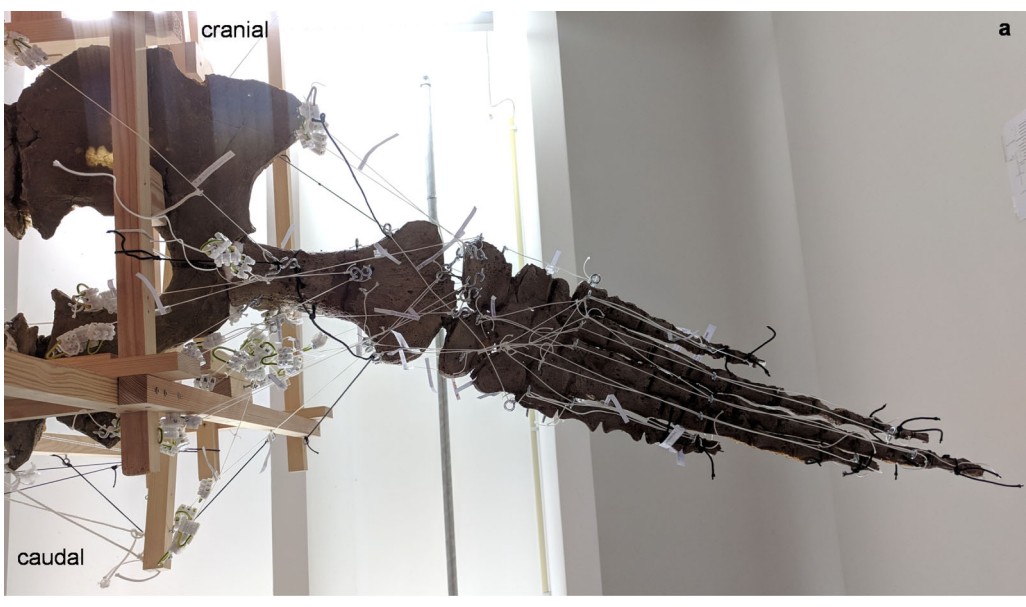

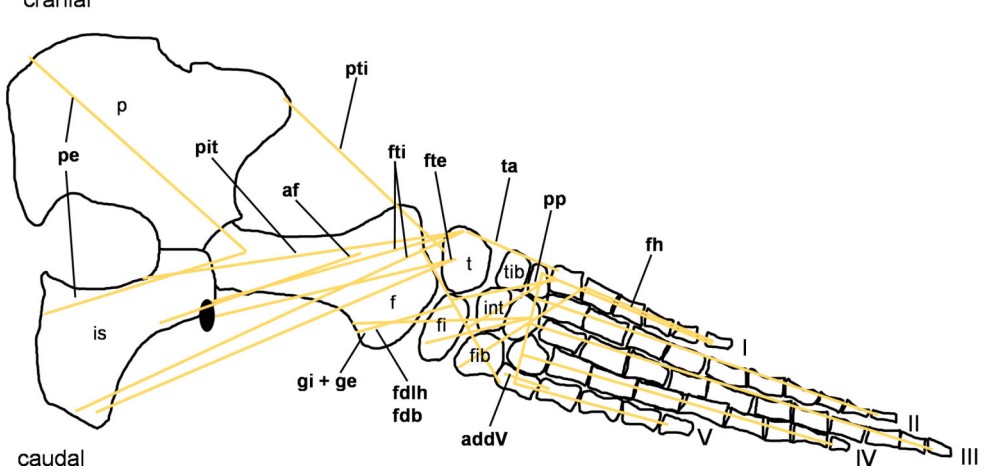

**Figure 5 Analog model of the myology of *Cryptoclidus eurymerus* (mounted skeleton IGPB R 324), pelvic girdle and hindflipper in ventral view.** (A) Mounted bone casts with white chords representing the lines of action of the hindflipper muscles. (B) Tracing of the pelvic girdle and hindflipper with muscle lines of action. Abbreviations of muscles: addV, musculus adductor digiti quinti; af, musculus adductor femoris; fdlh/fdb, musculus flexor digitorum longus (hindflipper)/musculi flexores digitores breves; fh, musculus flexor hallucis; fte, musculus flexor tibialis externus; fti, musculus flexor tibialis internus; gi and ge, musculus gastrocnemius internus and musculus gastrocnemius externus; pe, musculus pu-bo-ischiofemoralis externus; pit, musculus puboischiotibialis; pti, musculus pubotibialis; pp, musculus pronator profundus; ta, musculus tibialis anterior. Abbreviations of bones: f, femur; fi, fibula; fib, fibulare; int, intermedium; is, ischium; p, pubis; t, tibia; tib, tibiale; I, digit one; II, digit two; III, digit three; IV, digit four; V, digit five.

assumed force vectors for each muscle were closer, on average, to adjacent bone elements. These muscles may have been held close by annular pulleys. Terminology for the limb motion cycle follows *Rivera, Wyneken & Blob (2011)* and *Krahl & Witzel (2021)*, who use humeral and femoral depression, elevation, protraction, and retraction, which reflect well the requirements for underwater flight.

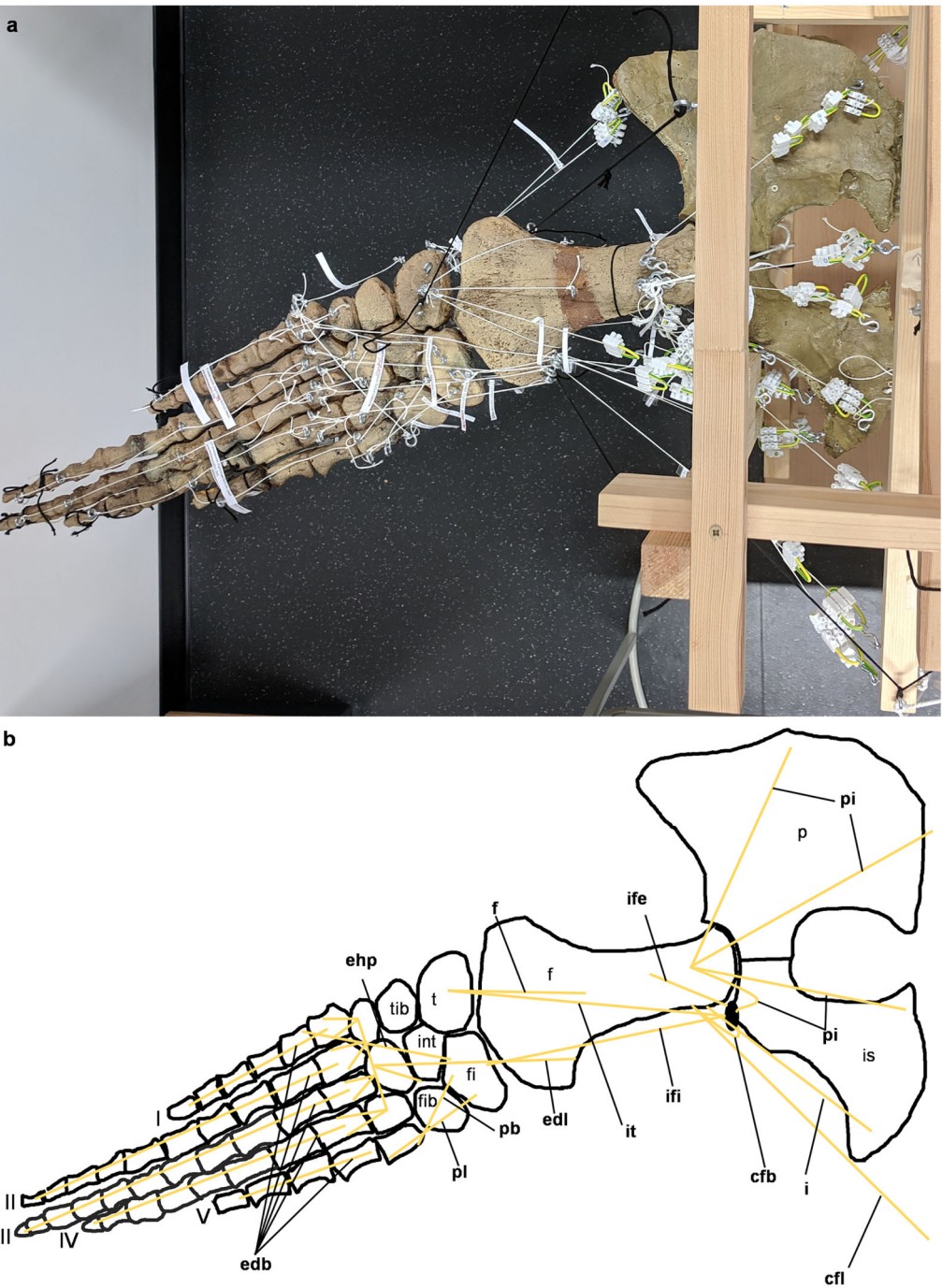

**Figure 6 Analog model of the myology of *Cryptoclidus eurymerus* (mounted skeleton IGPB R 324), pelvic girdle and hindflipper in dorsal view.** (A) Mounted bone casts with white chords representing the lines of action of the hindflipper muscles. (B) Tracing of the pelvic girdle and hindflipper with muscle lines of action. Abbreviations of muscles: cfb, musculus caudifemoralis brevis; cfl, musculus caudifemoralis longus; edb, musculi extensors digitores breves; edl, musculus extensor digitorum longus; ehp, musculus extensor hallucis proprius; f, musculus femorotibialis; ife, musculus iliofemoralis; ifi, musculus iliofibularis; it, musculus iliotibialis; pb, musculus peroneus brevis; pl, musculus peroneus longus; pi, musculus puboischiofemoralis internus. Abbreviations of bones: f, femur; fi, fibula; fib, fibulare; int, intermedium; is, ischium; p, pubis; t, tibia; tib, tibiale; I, digit 1; II, digit 2; III, digit 3; IV, digit 4; V, digit 5.

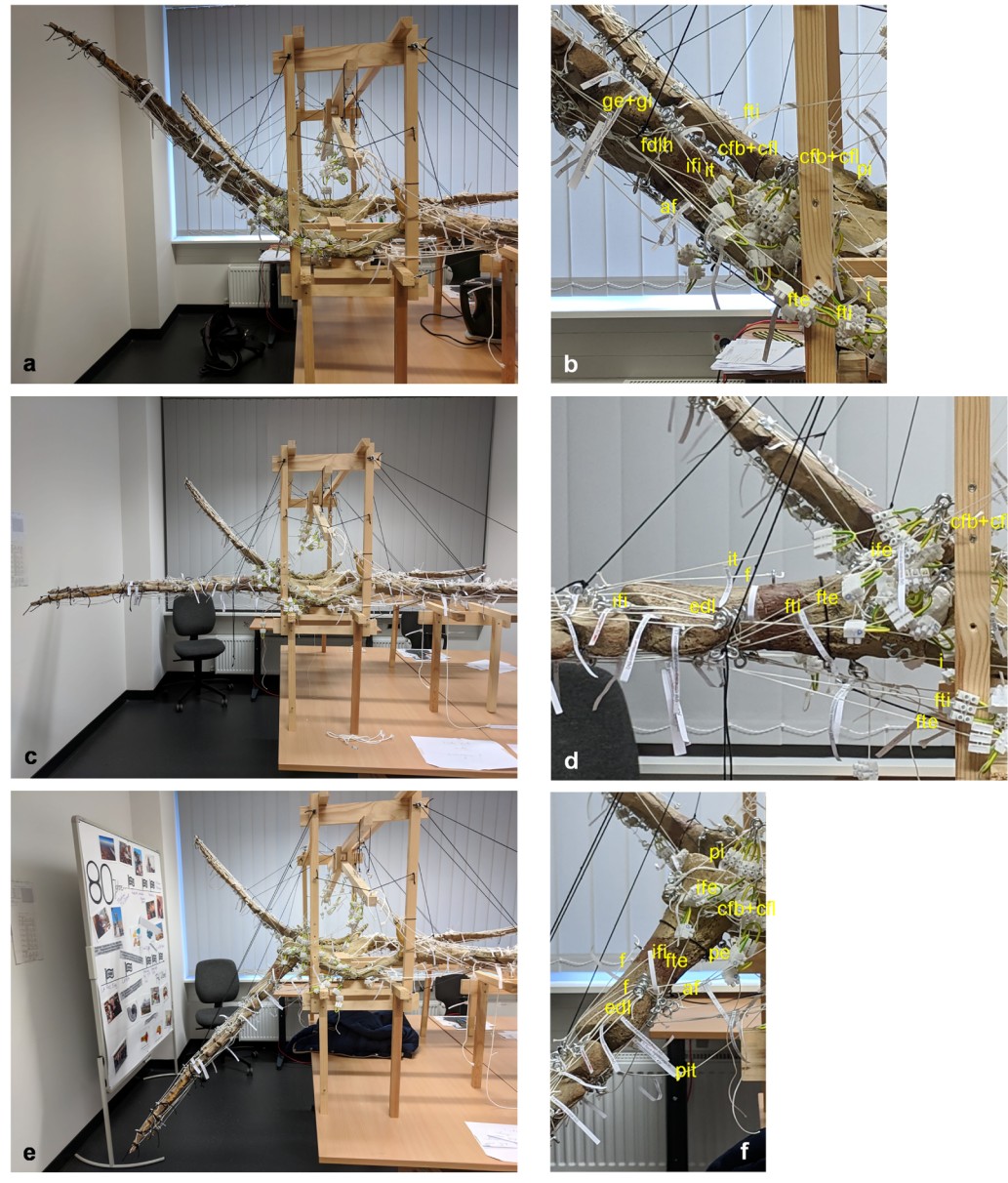

**Figure 7** **Analog model of the muscle lines of action of the hindflipper of *Cryptoclidus eurymerus* (IGPB R 324) in posterior and dorsal views.** The images in the left column (A, C, E) show an overview of the flipper muscle lines of action. The images in the right column (B, D, F) focus on the femur muscles. (A) and (B) hindflipper at maximum dorsal excursion in posterior view. (C) and (D) hindflipper in the neutral position in posterior view. (E) and (F) hindflipper at maximal ventral excursion in posterior view. Abbreviations: af, musculus adductor femoris; cfb + cfl, musculus caudifemoralis brevis + musculus caudifemoralis longus; edl, musculus extensor digitorum longus; f, musculus femorotibialis; fdlh, musculus flexor digitorum longus (hindflipper); fte, musculus flexor tibialis externus; fti, musculus flexor tibialis internus; gi + ge, musculus gastrocnemius internus and musculus gastrocnemius externus; i, musculus ischiotrochantericus; ife, musculus iliofemoralis; ifi, musculus iliofibularis; it, musculus iliotibialis; pe, musculus puboischiofemoralis externus; pi, musculus puboischiofemoralis internus; pit, musculus puboischiotibialis.       
**Table 1 Changes in length of *Cryptoclidus eurymerus* (IGPB R 324) humerus and femur muscles.**

| Musculus | Number coding for muscle (portion) in respective diagram | Maximum ventral excursion (cm) | Neutral position (cm) | Maximum dorsal excursion (cm) | Muscle length change (− = shortening, + = lengthening) from neutral position to maximum ventral excursion (cm) | Muscle length change (− = shortening, + = lengthening) from neutral position to maximum ventral excursion (%) | Muscle length change (− = shortening, + = lengthening) from neutral position to maximum dorsal excursion (cm) | Muscle length change (− = shortening, + = lengthening) from neutral position to maximum dorsal excursion (%) | Muscle stretching = muscle neutral position (=100%) + muscle lengthening (%) | Muscle contraction = muscle neutral position (=100%) - shortening of muscle (%) | Total length change of muscle (%) (= muscle stretching - muscle contraction) |
|---|---|---|---|---|---|---|---|---|---|---|---|
| m. deltoideus scapularis | 1 | 22.2 | 13.6 | 9.8 | 8.6 | 63.24 | −3.8 | −7.63 | 163.24 | 92. 37 | 70.87 |
| m. deltoideus scapularis corrected | 2 | 22.6 | 22.6 | 22.6 | 0 | 0 | 0 | 0 | 100 | 100 | 0 |
| m. deltoideus clavicularis | 3 | 36.5 | 36.5 | 36.5 | 0 | 0 | 0 | 0 | 100 | 100 | 0 |
| m. latissimus dorsi (anterior) | 4 | 42.7 | 34.9 | 29.8 | 7.8 | 22.35 | −5.1 | −14.61 | 122.35 | 85.39 | 36.96 |
| m. latissimus dorsi (in between) | 5 | 43.9 | 37.6 | 32.6 | 6.3 | 16.76 | −5 | −13.3 | 116.76 | 86.7 | 30.06 |
| m. latissimus dorsi (posterior) | 6 | 52.7 | 47.9 | 42.2 | 4.8 | 10.02 | −5.7 | −11.9 | 110.02 | 88.1 | 21.92 |
| m. subcoracoscapularis (anterior portion) | 7 | 32.8 | 29.3 | 24.3 | 3.5 | 11.95 | −5 | −17.06 | 111.95 | 82.94 | 29.01 |
| m. subcoracoscapularis (posterior portion) | 8 | 36.3 | 33.2 | 33.2 | 3.1 | 9.34 | 0 | 0 | 109.34 | 100 | 9.34 |
| m. scapulohumeralis anterior | 9 | 30.9 | 29.5 | 27.4 | 1.4 | 4.75 | −2.1 | −7.12 | 104.75 | 92.88 | 11.87 |
| m. scapulohumeralis posterior | 10 | 24.9 | 22.7 | 19.4 | 2.2 | 9.69 | −3.3 | −14.54 | 109.69 | 85.46 | 24.23 |
| m. coracobrachialis brevis (anterior) | 11 | 29.3 | 36.7 | 40.5 | −7.4 | −20.16 | 3.8 | 10.35 | 110.35 | 79.84 | 30.51 |
| m. coracobrachialis brevis (posterior) | 12 | 43.8 | 43.8 | 45.5 | 0 | 0 | 1.7 | 3.88 | 103.88 | 100 | 3.88 |
| m. coracobrachialis longus | 13 | 43.1 | 52 | 53.9 | −8.9 | −17.12 | 1.9 | 3.65 | 103.65 | 82.88 | 20.77 |
| m. pectoralis (anterior) | 14 | 29.9 | 30.4 | 35.6 | −0.5 | −1.64 | 5.2 | 17.11 | 117. 11 | 98.36 | 18.75 |
| m. pectoralis (posterior) | 15 | 27.4 | 33.1 | 39.2 | −5.7 | −17.22 | 6.1 | 18.43 | 118.43 | 82.78 | 35.65 |
| m. supracoracoideus | 16 | 24.5 | 31.7 | 31.7 | −7.2 | −22.71 | 0 | 0 | 100 | 77.29 | 22.71 |
| m. biceps brachii | 17 | 56.7 | 67.8 | 72.4 | −11.1 | −16.37 | 4.6 | 6.78 | 106.78 | 83.63 | 23.15 |
| m. triceps brachii (anterior) | 18 | 36.8 | 36.8 | 36.8 | 0 | 0 | 0 | 0 | 100 | 100 | 0 |
| m. triceps brachii (posterior) | 19 | 31.2 | 31.2 | 31.2 | 0 | 0 | 0 | 0 | 100 | 100 | 0 |
| m. caudofemoralis longus | 1 | 61.3 | 63.5 | 64.6 | −2.2 | −3.46 | 1.1 | 6.15 | 106.15 | 96.54 | 9.61 |

| Musculus | Number coding for muscle (portion) in respective diagram | Maximum ventral excursion (cm) | Neutral position (cm) | Maximum dorsal excursion (cm) | Muscle length change (− = shortening, + = lengthening) from neutral position to maximum ventral excursion (cm) | Muscle length change (− = shortening, + = lengthening) from neutral position to maximum ventral excursion (%) | Muscle length change (− = shortening, + = lengthening) from neutral position to maximum dorsal excursion (cm) | Muscle length change (− = shortening, + = lengthening) from neutral position to maximum dorsal excursion (%) | Muscle stretching = muscle neutral position (=100%) + muscle lengthening (%) | Muscle contraction = muscle neutral position (=100%) - shortening of muscle (%) | Total length change of muscle (%) (= muscle stretching - muscle contraction) |
|---|---|---|---|---|---|---|---|---|---|---|---|
| m. caudofemoralis brevis (ilium) | 2 | 17.9 | 17.9 | 17.9 | 0 | 0 | 0 | 0 | 100 | 100 | 0 |
| m. caudofemoralis brevis (vc) | 3 | 37.6 | 36.9 | 33.6 | 0.7 | 1.9 | −3.3 | −8.94 | 101.9 | 96.7 | 5.2 |
| m. flexor tibialis internus (vc) | 4 | 63.7 | 58.4 | 52.3 | 5.3 | 9.8 | −6.1 | −10.45 | 109.8 | 93.9 | 15.9 |
| m. flexor tibialis internus (ischium) | 5 | 43.1 | 47.1 | 49.2 | −4 | −8.49 | 2.1 | 4.46 | 104.46 | 91.51 | 12.95 |
| m. flexor tibialis externus (ischium) | 6 | 42.8 | 47.5 | 49.4 | −4.7 | −9.89 | 1.9 | 4 | 104 | 90.11 | 13.89 |
| m. flexor tibialis externus (ilium) | 7 | 42.6 | 37.2 | 31.9 | 5.4 | 14.52 | −5.3 | −14.25 | 114.52 | 85.75 | 28.77 |
| m. iliofibularis | 8 | 33.1 | 29 | 25.3 | 4.1 | 14.13 | −3.7 | −12.76 | 114.13 | 87.24 | 26.89 |
| m. ambiens | 9 | 29.3 | 29.3 | 31.7 | 0 | 0 | 2.4 | 8.19 | 108.19 | 100 | 8.19 |
| m. puboischiotibialis | 10 | 38.3 | 42.3 | 45.8 | −4 | −9.46 | 3.5 | 8.27 | 108.27 | 90.54 | 17.73 |
| m. pubotibialis | 11 | 33.4 | 33.4 | 33.4 | 0 | 0 | 0 | 0 | 100 | 100 | 0 |
| m. iliofemoralis | 12 | 18.3 | 14.2 | 13.6 | 4.1 | 28.87 | −0.6 | −4.23 | 128.87 | 95.77 | 33.1 |
| m. ischiotrochantericus | 13 | 21.5 | 21.5 | 23.7 | 0 | 0 | 2.2 | 10.23 | 110.23 | 100 | 10.32 |
| m. iliotibialis | 14 | 40.1 | 26.6 | 26.6 | 7.3 | 27.44 | 0 | 0 | 127.44 | 100 | 27.44 |
| m. adductor femoris (anterior) | 15 | 30.4 | 32.8 | 36.2 | −2.4 | −7.31 | 3.4 | 6.71 | 106.71 | 92.69 | 14.02 |
| m. adductor femoris (lateroposterior) | 16 | 26.6 | 30 | 32.9 | −3.4 | −11.33 | 2.9 | 9.67 | 109.67 | 88.67 | 21 |
| m. puboischiofemoralis internus (pubis) | 17 | 36.8 | 32.3 | 30.3 | 4.5 | 14.85 | −2 | −6.19 | 114.85 | 93.81 | 21.04 |
| m. puboischiofemoralis internus (ischium) | 18 | 29.4 | 27.2 | 23.3 | 2.2 | 8.09 | −3.9 | −14.34 | 108.09 | 85.66 | 22.43 |
| m. puboischiofemoralis internus (ilium) | 19 | 10.4 | 9.5 | 9.5 | 0.9 | 9.47 | 0 | 0 | 109.47 | 100 | 9.47 |
| m. puboischiofemoralis internus (vertebral column) | 20 | 39.6 | 32.1 | 28.1 | 7.5 | 23.36 | −4 | −12.46 | 123.36 | 87.54 | 35.82 |
| m. puboischiofemoralis externus (pubis) | 21 | 39.1 | 42.5 | 45.7 | −3.4 | −8 | 3.2 | 7.5 | 107.5 | 92 | 15.5 |
| m. puboischiofemoralis externus (ischium) | 22 | 22.9 | 27.6 | 29.4 | −4.7 | −17.03 | 1.8 | 6.52 | 106.52 | 82.97 | 23.55 |

**Table 2 Agonistic and antagonistic humerus muscles of *Cryptoclidus eurymerus* (IGPB R 324).**

| Agonists | Antagonists |
|---|---|
| anterior portion of m. latissimus dorsi (eventually m. scapulohumeralis posterior and m. scapulohumeralis anterior) (elevation, protraction) | posterior portion m. pectoralis (depression and retraction) |
| posterior portion of m. latissimus dorsi (elevation, retraction) | anterior portion of m. pectoralis (protraction, depression) |
| m. subcoracoscapularis (anterior portion), m. deltoideus scapularis (both elevation, protraction) | m. coracobrachialis longus, m. coracobrachialis brevis, m. biceps, posterior portion of m. supracoracoideus (all retraction, depression) |
| m. subcoracoscapularis (posterior portion) (elevation, retraction) | anterior portion of m. supracoracoideus, m. deltoideus clavicularis (all depression, protraction) |
| m. latissimus dorsi, anterior portion of m. pectoralis, posterior portion of m. subcoracoscapularis, m. deltoideus scapularis, m. coracobrachialis brevis, m. coracobrachialis longus (rotation (leading edge upwards) | m. scapulohumeralis anterior, m. scapulohumeralis posterior, anterior portion of m. subcoracoscapularis, m. deltoideus clavicularis, posterior portion of m. pectoralis, m. biceps brachii, m. triceps brachii (leading edge downwards) |
| m. biceps (retraction, depression) | m. triceps (elevation, protraction) |
| m. extensor digitorum communis (extension metacarpals) | m. flexor digitorum longus (flexes digit I–V) |
| humeral triceps head (offsets ulna slightly dorsally), m. extensor carpi ulnaris (offsets ulna dorsally, or eventually extends metacarpal V) | m. flexor carpi ulnaris (displaces ulnar side of carpus ventrally, eventually flexes metacarpal V) |
| m. supinator longus and extensor carpi radialis (offsets radius or eventually the radial carpal side dorsally) | m. flexor carpi radialis (flexes metacarpal 1 or offsets the radial carpal side ventrally), m. pronator teres (offsets radius ventrally), m. brachialis (offsets radius slightly ventrally) |

**Table 3 Agonistic and antagonistic femur muscles of *Cryptoclidus eurymerus* (IGPB R 324).**

| Agonists | Antagonists |
|---|---|
| m. puboischiofemoralis internus (pubis, vertebral column) (protraction, elevation) | m. puboischiofemoralis externus (ischium), m. adductor femoris, m. flexor tibialis internus (ischium), m. flexor tibialis externus (ischium), m. ischiotrochantericus, m. puboischiotibialis (retraction, depression) |
| m. puboischiofemoralis externus (pubis, anterior) (protraction, and depression) | m. puboischiofemoralis internus (ischium, ilium), m. caudifemoralis brevis and m. caudifemoralis longus, m. iliofibularis, m. iliotibialis, m. iliofemoralis, m. flexor tibialis externus (ilium), m. flexor tibialis internus (vertebral column) (retraction elevation) |
| m. ambiens (protraction), m. pubotibialis (protraction) | m. iliofibularis (elevation, retraction) |
| m. puboischiofemoralis externus (pubis), m. puboischiofemoralis internus (ischium, ilium), iliofemoralis, iliotibialis (rotates flipper leading edge up), m. ambiens and m. pubotibialis (rotates flipper leading edge up, if tibia below origin area) | m. puboischiofemoralis externus (ischium), m. puboischiofemoralis internus (pubis), m. adductor femoris, m. ischiotrochantericus, m. flexor tibialis internus, m. caudifemoralis brevis, m. caudifemoralis longus, m. flexor tibialis externus, puboischiotibialis, m. iliofibularis (rotates flipper leading edge down), m. ambiens and m. pubotibialis (rotates flipper leading edge down, if tibia above origin area) |
| m. extensor digitorum longus (digital extensor), femorotibialis (offsets tibia dorsally) | m. gastrocnemius internus + m. gastrocnemius externus, m. flexor digitorum longus (digital flexors) |

## Determination of changes in flipper muscle length

Total muscle length changes were measured on the analog model to determine if the experimental data could be used to reconstruct plesiosaur muscles. Three positions in the inferred beat cycle of the foreflipper and hindflipper were selected for determining the total muscle length change of muscles originating at the pectoral and pelvic girdles or vertebrae and inserting or spanning the humerus and femur. These flipper positions are

the maximum dorsal excursion of the fore- and hindflipper during upstroke (~+50° from horizontal) (Figs. 4A, 4B, 7A and 7B), the maximum ventral excursion of the fore- and hindflipper during downstroke (~−50°) (Figs. 4E, 4F, 7E and 7F), and the neutral position (0°) (Figs. 4C, 4D, 7C and 7D). Based on glenoid and acetabular alignment, the tip of the foreflipper points slightly anteriorly, and the tip of the hindflipper points slightly posteriorly. Protraction and retraction were not considered because they contribute little to the flipper beat cycle inferred from osteology. The degrees of maximum elevation and depression of the flipper were based on the determination of flipper motion by *Carpenter et al. (2010)* and *Liu et al. (2015)* for several plesiosaur species. However, we note that the inferences on degree of excursion above and below the horizontal are highly dependent on how much cartilage presumably covered the humerus and femur, as well as the glenoid and acetabulum, as noted by *Liu et al. (2015)*, and on plesiosaur species (which differ in the size of the dorsal tuberosity/trochanter).

We chose not to refer to the neutral flipper position as the resting position of the flipper because it is probably impossible to determine the resting position of the flipper for an extinct species. Furthermore, in the muscle physiology literature, the term "resting length" usually refers to individual sarcomeres (*Rassier, MacIntosh & Herzog, 1999*) or fascicles of a muscle (*Biewener & Roberts, 2000*). Furthermore, not knowing the exact flipper muscle resting length affects the exact value of muscle stretching and contraction, but not the final result of the total length change of a muscle, which was calculated in this study (see below). This is because the absolute values of muscle stretching and contraction are indeed measured with respect to the neutral position, but the total change in length of the muscle is the difference between the maximum muscle stretching and contraction.

All three positions of the fore- and hindflippers were fixed in sequence with black nylon cords suspending the flippers from the wooden frame. For each of the three positions, the length of each muscle was fixed using the terminal strips by tightening the small screws in the terminal strips on the muscle cords. Then, each muscle cord was removed from the model by unhooking it from the screw eye pins, and all three muscle lengths, *i.e.*, maximum deflection on downstroke, neutral position, and maximum deflection on upstroke, were measured in cm using a tape measure as the distance between the respective terminal strips. The changes in length between the maximum deflection on the downstroke and the neutral position and between the maximum deflection on the upstroke and the neutral position were recalculated as percentages, with the resting length set to 100%. Then, muscle stretching, muscle contraction, and the difference between the two, the total change in muscle length, were calculated in % (Table 1). Bar graphs for total muscle length change in % were created using Microsoft Excel (Figs. 8A and 8B). Some muscle length changes were given as 0 cm because the actual length changes could not be measured because they were smaller than the width of the terminal strips. In one case, *i.e.*, the m. deltoideus scapularis, the total length change of the muscle was found to be unphysiological when attached to the lateral on the scapula blade (see below). Hence, a different area of origin (at the ventrolateral part of the scapula) was tested, which yielded physiologically plausible results, which were then measured for all three positions.

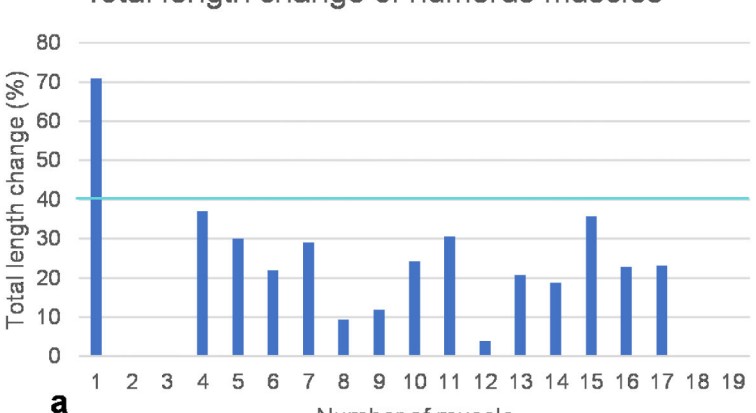

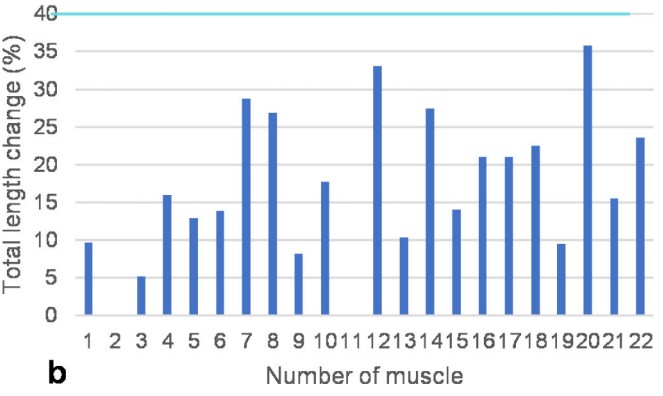

**Figure 8 Total length change of muscles.** (A) Humeral muscles and their parts. (B) Femoral muscles and their parts. Note that the locomotor muscles attaching to or originating from the humerus and femur cover the entire spectrum of maximum total length change of vertebrate muscles, ranging from non-measurable total length change for muscles with a complex architecture to about 40% of total length change typical for approximately parallel-fibered muscles. The turquoise line indicates a length change of 40% marking the approximate physiological limit (*Biewener, Corning & Tobalske, 1998*; *Biewener & Roberts, 2000*). Note also that the m. deltoideus scapularis (1) of the humerus would have had a non-physiological length change of >70% if it would have originated from the lateral scapular blade. If it would have originated from the ventral scapula (2), its total length change is within the physiological range and below the measurement error (*i.e.*, typical for muscles with a complex internal architecture). Muscle numbers are found in Table 1.           

## FESA and computational determination of muscle strength

FESA requires a 3D model of the bones to be investigated. Hence, the right humerus and left femur were removed from the mounted skeleton IGPB R 324 and scanned with the micro-CT scanner at IGPB. The scanner is an industrial high-resolution computed tomography scanner (model Phoenix v|tome|x s 240, manufactured by General Electric Phoenix X-ray, Wunstorf, Germany). The scans were processed using the dedicated datos| x software and the VGStudio MAX (Volume Graphics) program to obtain image stacks in the *z*-direction for the humerus and femur.

The two image stacks were loaded into Simpleware ScanIP 5.1 (*Krahl et al., 2019*) for further processing which consisted of segmentation and generating the 3D models. For each image in the stack, the bone surface was selected and segmented using grayscale intervals, and a 3D surface model was created and saved in .stl file format (Data S1 and Data S2). We did not model the internal inhomogeneities in the bone, *i.e.*, cortical *vs.* spongy, because this would have precluded implementation of torsion in the FESA. The reason is that the much stiffer cortex would take up all the stress peaks.

The 3D model was then imported into ANSYS 16.0 (ANSYS Inc., Canonsberg, PA, USA) as the .stl file. The dimensions of the humerus and femur from IGPB R 324 were used to scale the respective volumetric model to the original size. Subsequently, the proximal articular cartilage or articular surface between the glenoid and acetabulum was modeled as a volume. Bonded contacts between the artificial cartilage structures were implemented in the analysis. The models were constrained by fixed boundaries on the upper surface of the cartilage volume. Similar to other established finite element models (*Gil, Marcé-Nogué & Sánchez, 2015*; *Sellés de Lucas et al., 2018*), the material was assumed

to be linear-elastic and isotropic to reduce computational time. The FE models were meshed with 10-node tetrahedral elements (SOLID92). The humerus model consisted of 92665 elements, and the femur model consists of 75,784 elements. In the humerus, the cartilaginous joint structure is formed by 19,927 elements and in the femur by 15,472 elements. In both FE models, the bone was modeled with a Young's modulus of 12,000 MPa (*Currey, 1987*) and the cartilage with a Young's modulus of 5 MPa (*Carter & Beaupré, 2001*) and a Poisson's ratio of 0.3 for both materials.

Next, the LOA and muscle insertions had to be implemented in the FE model. We did this by taking photographs of the analog model in anterior, posterior, dorsal and ventral views and in these traced the LOA and their angles of attachment on the humerus (Fig. 9A) and femur (Fig. 10E). We then implemented these angles of attachment in the FE models as vectors from two views of the respective angle of attachment (Figs. 9B and 10B). Two- or more-headed muscle bellies were implemented in the FE model in the form of the resultant vector. The two-joint muscles of the foreflipper are the m. triceps brachii and the m. biceps brachii, whereas the two-joint muscles of the hindflipper are m. ambiens, m. pubotibialis, m. flexor tibialis externus, m. flexor tibialis internus, m. iliotibialis, m. iliofibularis, and m. puboischiotibialis. Since it is not possible to represent curved vectors in ANSYS 16.0., the muscle wraps were modeled by dividing their lines of action into several smaller straight vectors.

Force transmission to the distal articular surfaces of the humerus and femur due to muscle activity was initially applied in one point. This resulted in artificially high, localized stress peaks. Therefore, the force application was divided into several application points distributed over the distal articular surfaces to perform a more realistic simulation of load application over one surface.

The stress distribution was calculated for both bones (Figs. 9 and 10C–10N). The forces of each humerus and femur muscle optimized using the ANSYS subproblem approximation method. Briefly, in this method, the first and third principal stresses were analyzed for 12 different nodes that were uniformly distributed over the outer bone surface and served as state variables. In addition, the deformation of the model was evaluated. The initial estimated starting points for the muscle force calculations were based on the general force level differences and correlations reported by *Krahl et al. (2019)* for a sea turtle. Optimization criteria were as follows: Minimization of displacement at a static time step, first principal stresses between 0 and 0.5 MPa, third principal stresses between −10 and −0.1 MPa. For each model and load case, 40 iterations were performed. The stress diagrams were then visually inspected, and some final adjustments were made manually. In this way, the muscle forces were iteratively approximated (Tables 3 and 4) (*Witzel & Preuschoft, 2005*; *Sverdlova & Witzel, 2010*). For the final analysis, the principal stresses were saved, and the 1st and 3rd principal stresses were plotted. In addition, the shear stress was calculated according to *Carter & Beaupré (2001)*.

## Load case generation for FESA
Two load cases, downstroke and upstroke, were selected for calculation to reflect the constantly changing load on the humerus and femur during the flipper beat cycle (Figs.

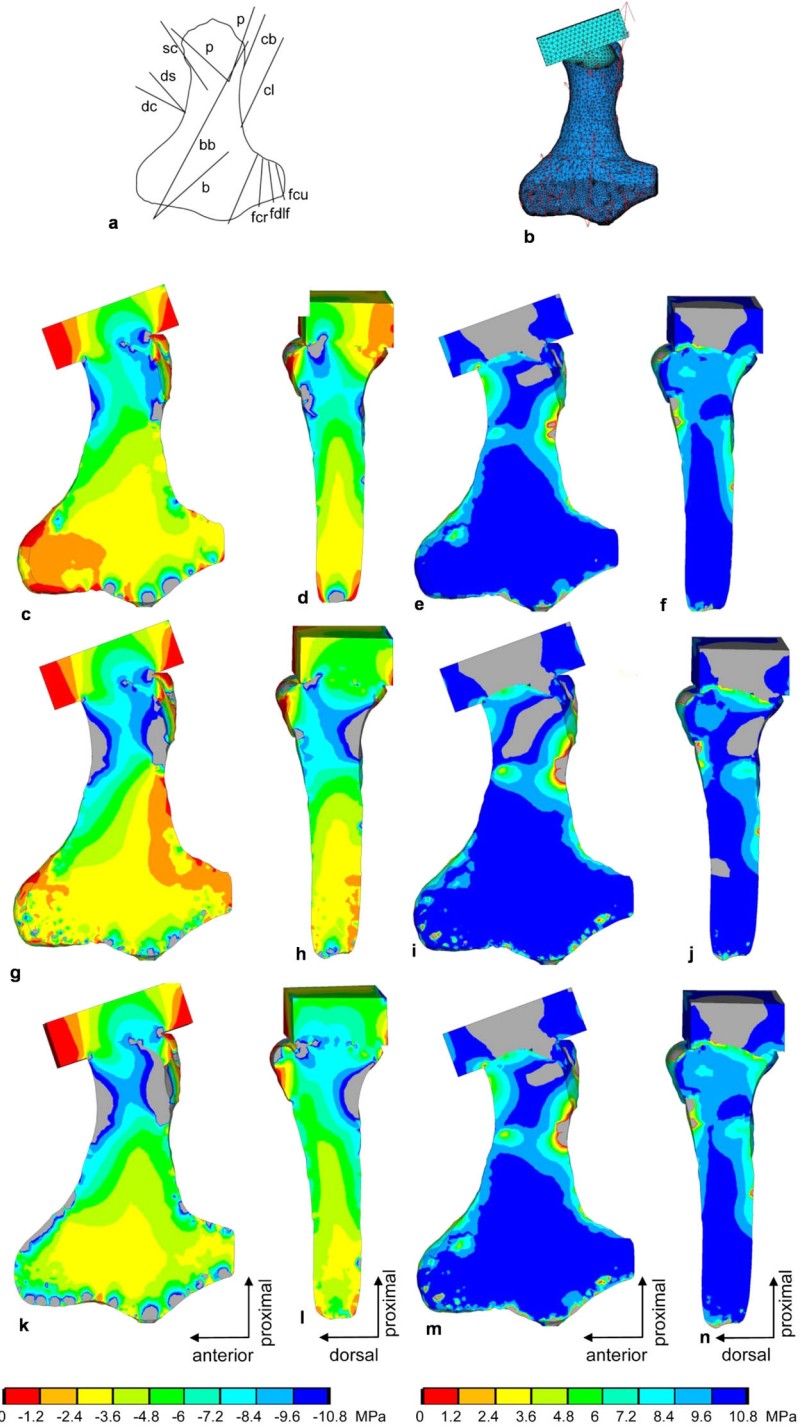

**Figure 9 FESA of the humerus of *Cryptoclidus eurymerus* (IGPB R 324).** (A) Outline drawing of the humerus and lines of action of the muscles attaching to it as obtained from the myological analysis. (B) Mesh volumetric FE model with force vectors of attaching muscles (red arrows). (C–F) FESA of load case 1 (downstroke). (G–J) FESA of load case 2 (upstroke). (K–N) FESA superposition of both load cases 1 and 2. The left two columns (C), (D), (G), (H), (K), (L) show the compressive stress distribution, and the right two columns (E), (F), (I), (J), (M), (N) show the tensile stress distribution. The color spectrum indicates the compressive and tensile stress in MPa. Note that the areas of lower compressive stress correspond to the areas of cancellous bone (on the inside of the bone) and the areas of higher compressive
**Figure 9** (continued)
stress correspond to cortical bone (on the outside of the bone). Abbreviations: b, musculus brachialis bb, musculus biceps brachii; cb, musculus coracobrachialis brevis; cl, musculus coracobrachialis longus; dc, musculus deltoideus clavicularis; ds, musculus deltoideus scapularis; fcr, musculus flexor carpi radialis; fcu, musculus flexor carpi ulnaris; fdlf, musculus flexor digitorum longus (foreflipper); p, musculus pectoralis; pte, musculus pronator teres; sc, musculus supracoracoideus.

9C–9J and 10C–10J). The two load cases were later superimposed (Figs. 9K–9N and 10K–10N) to obtain a model functionally loaded by compression only. This is because the biological response of bone to mechanical stimuli is time-dependent. To calculate the long-term functional mechanical load, a physiological superposition of all stresses is performed: For each finite element, the highest principal stresses occurring in both load cases are summarized (*Witzel & Preuschoft, 2005*).

For both load cases, a position was chosen in which the humerus was held horizontally at the level of the glenoid and pointed laterally and slightly forward, as in the analog model. Similarly, the femur was positioned horizontally at the level of the acetabulum with the flipper tip pointing mainly laterally, but also angled slightly posteriorly. On the downstroke, the humerus and femur were also rotated anteriorly downward around their long axis by approximately 19° (*Witzel, Krahl & Sander, 2015*; *Witzel, 2020*) to model a position where the flippers would have generated lift. During the upstroke, the humerus and femur were rotated posteriorly around their long axes and downward by approximately 19°. We justify these angles of rotation as follows (*Witzel, Krahl & Sander, 2015*; *Witzel, 2020*): Assuming a flight speed of 2 m/s and a flipper stroke speed of 4 m/s at the distal flipper tips (based on *Robinson (1975)*), a rotation angle of about 19° is necessary to minimize angle-dependent drag and optimize lift generation (and thus underwater flight).

For the implementation of load cases, it is crucial to identify and consider which muscles act as agonists and antagonists (*Witzel & Preuschoft, 2005*). The shoulder and hip joint may allow a small amount of play for rotation and protraction/retraction and a large amount of play for elevation and depression of the flippers, which is stabilized by the interaction of agonists and antagonists.

To identify agonists and antagonists, opposing muscle functions (Tables 2 and 3) are taken from *Krahl & Witzel (2021)*. The downstroke is driven primarily by the humeral and femoral depressors, but also by retractors and those muscles that allow slight downward rotation of the flipper leading edges. The flexor muscles of the humerus and femur are active during the downstroke, flexing the digits and contributing to the twisting of the fore- and hindflipper along the flipper lengths. The upstroke is largely driven by the humeral and femoral elevators. Protraction of the humerus and femur and upward rotation of the leading edge of the flipper also contribute to the upstroke. The extensor muscles originating from the distal humerus and femur assist in twisting the flipper and extending the digits during the upstroke (*Krahl, 2021*).

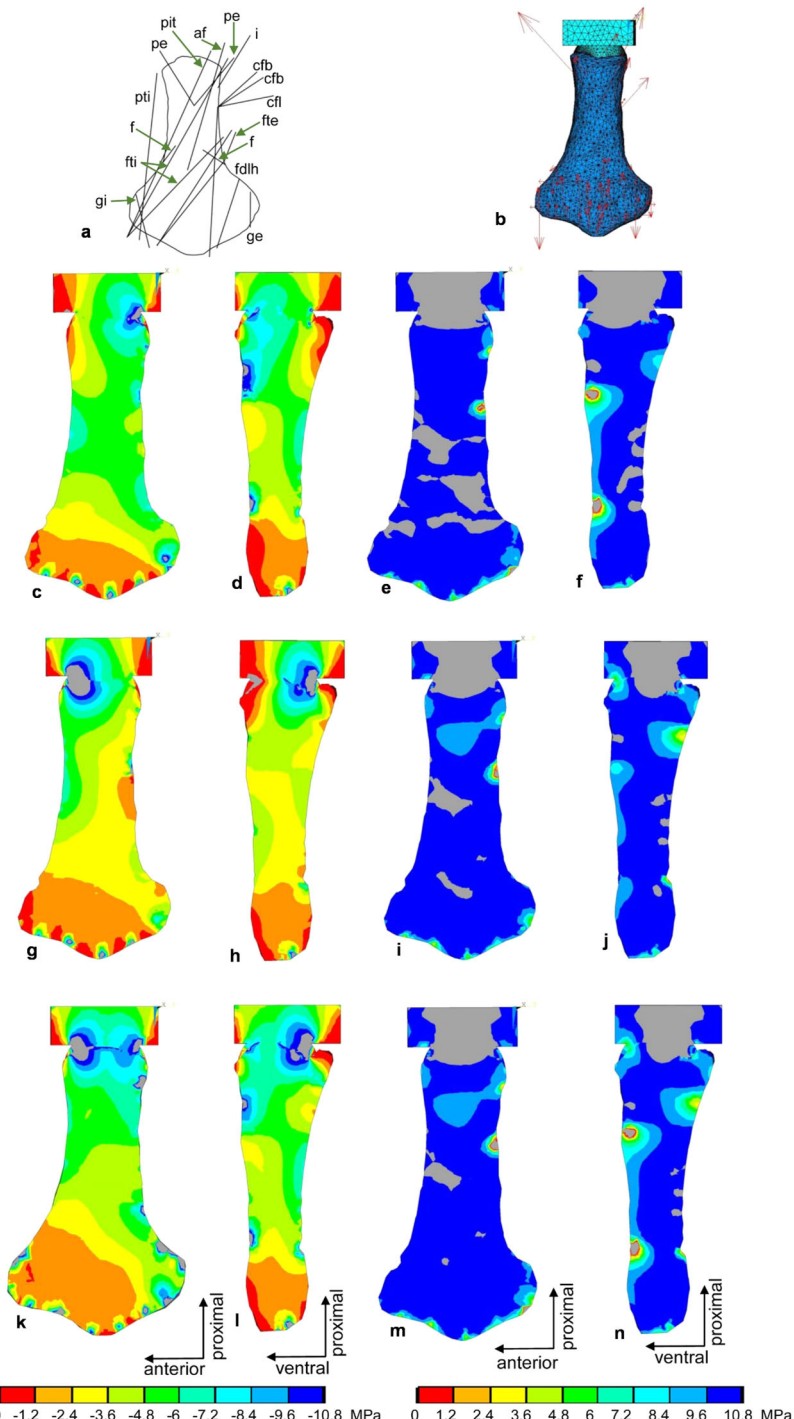

**Figure 10 FESA of the femur of *Cryptoclidus eurymerus* (IGPB R 324).** (A) Outline drawing of the femur and lines of action of the muscles attaching to it to it as obtained from the myological analysis. (B) Mesh volumetric FE model with force vectors of attaching muscles (red arrows). (C–F) FESA of load case 1 (downstroke). (G–J) FESA of load case 2 (upstroke). (K–N) FESA superposition of both load cases 1 and 2. The left two columns (C), (D), (G), (H), (K), (L) show the compressive stress distribution, and the right two columns (E), (F), (I), (J), (M), (N) show the tensile stress distribution. The color spectrum indicates the compressive and tensile stress in MPa. Note that the areas of lower compressive stress correspond to the areas of cancellous bone on the inside of the bone and the areas of higher compressive

Figure 10 (continued)
 stress correspond to cortical bone on the outside of the bone. Abbreviations: af, musculus adductor femoris; cfb, musculus caudifemoralis brevis; cfl, musculus caudifemoralis longus; f, musculus femorotibialis; fdlh, musculus flexor digitorum longus; fte, musculus flexor tibialis externus; fti, musculus flexor tibialis internus; gi and ge, musculus gastrocnemius internus and musculus gastrocnemius externus; i, musculus ischiotrochantericus; pe, musculus puboischiofemoralis externus; pit, musculus puboischiotibialis; pti, musculus pubotibialis.

Table 4 Muscle forces of *Cryptoclidus eurymerus* (IGPB R 324) humerus by superposition of FESA load cases.

| Muscle | Muscle force (N) |
| --- | --- |
| m. supracoracoideus | 6,000 |
| m. coracobrachialis brevis | 4,800 |
| m. coracobrachialis longus | 3,600 |
| m. deltoideus clavicularis | 1,500 |
| m. deltoideus scapularis | 1,649 |
| m. scapulohumeralis anterior | 2,400 |
| m. scapulohumeralis posterior | 1,920 |
| m. brachialis | 324 |
| m. triceps humeral head | 275 |
| m. pectoralis | 9,600 |
| m. subcoracoscapularis | 4,422 |
| m. latissimus dorsi | 3,918 |
| m. extensor carpi ulnaris | 1,000 |
| m. extensor digitorum communis | 6,000 |
| m. extensor carpi radialis | 1,000 |
| m. pronator teres | 640 |
| m. flexor carpi ulnaris | 3,000 |
| m. flexor digitorum longus | 1,500 |
| m. flexor carpi radialis | 1500 |

# RESULTS

## Tests of plesiosaur flipper muscle reconstructions
### Implications of FESA for foreflipper musculature reconstruction
As detailed above, humeral muscles reconstructed based on comparative anatomy (*Krahl & Witzel, 2021*) were tested for functionality on a mechanical basis. Based on the FESA, certain possible muscle attachment areas are functionally preferable over others. These authors reconstructed the m. deltoideus scapularis on the anteroventral scapula posterior to the m. deltoideus clavicularis and anterior to the m. supracoracoideus, as well as on the lateral median scapular blade. Here, we can safely rule out an origin of the m. deltoideus scapularis on the scapula, as a total muscle length change of 70% indicates extreme muscle shortening, which is physiologically impossible (see below) (Fig. 8A; Table 1).

Myologic reconstruction (*Krahl & Witzel, 2021*) suggests an origin of the m. biceps brachii at the posterior ventral region of the coracoid. The m. biceps brachii and m. brachialis could have inserted either on the proximal radius *via* a common tendon or on the posteroproximal part of the radius and anteroproximal part of the ulna. In addition, the m. triceps brachii was suggested to originate anterodorsally from the bony ridge surrounding the glenoid on the scapula and from the coracoid immediately posterior to the glenoid and suggested to insert into the posterodorsal ulna (*Krahl & Witzel, 2021*). The posterodorsal ulnar insertion of the m. triceps brachii is unquestionable based on the EPB, and the posterior origin of the m. biceps brachii is also fairly well established. In contrast, the areas of origin of the m. triceps brachii are unclear, as is the insertion area of the m. biceps brachii and m. brachialis. The FESA results were improved *i.e.*, achieved a more homogeneous compressive stress distribution, when the attachment of the m. biceps brachii/m. brachialis was, mechanically advantageous, at the proximal and ventral radius and when the origin of the m. triceps brachii anteriorly to anterodorsally to the glenoid facet was pronounced by a higher muscle force. An origin of the m. triceps brachii on the coracoid would have provided little to no leverage and therefore may have been reduced or lost.

Myological reconstruction based on osteological correlates (*Krahl & Witzel, 2021*) suggests insertion of the m. coracobrachialis brevis and m. coracobrachialis longus at the posterior to posteroventral humeral shaft, as observed in lepidosaurs (*Russell & Bauer, 2008*). This reconstruction is not well supported by the EPB, which rather had suggested an insertion ventrally at the intertrochanteric fossa, as observed in turtles and crocodylians (*Walker, 1973*; *Meers, 2003*; *Suzuki & Hayashi, 2010*). Shifting the insertions further distally along the humerus shaft is mechanically advantageous because it increases the lever arms as can be observed well in the analogous model (Figs. 2A and 2B), and has been comparatively observed in Cheloniidae (*Walker, 1973*; *Krahl et al., 2019*).

The m. scapulohumeralis anterior was not reconstructed based on comparative myology by *Krahl & Witzel (2021)*, as this muscle is found only in lepidosaurs and has no synonyms (*Russell & Bauer, 2008*). Reconstructing it in plesiosaurs thus is a unipolar inference (*Snively & Russell, 2007*). Nevertheless, we formulate a description and a reconstruction of this muscle below, as it adds a necessary proximal rotational component to the musculature moving the humerus. Musculus scapulohumeralis anterior has two portions originating from the anterior ventrolateral scapula and from the posterolateral scapula dorsal to the glenoid, as seen in *Varanus exanthematicus* and *Iguana iguana* (*Jenkins & Goslow, 1983*; *Russell & Bauer, 2008*). In plesiosaurs, we now reconstruct it to originate at the anterior margin of the lateral scapula, at the base of the scapula, and at the posterior margin of the scapula. The lines of action favor the latter origin because the former would lead to a wrapping of the m. scapulohumeralis anterior around the lateral scapula and around the muscles that suspend the pectoral girdle from the trunk, which seems rather unlikely. However, an origin at the posterior scapular blade would also only be able to support a very small muscle belly, as the scapular blade of *Cryptoclidus* is generally very reduced compared to archosaurs, lepidosaurs, and turtles (*Walker, 1973*; *Meers, 2003*; *Russell & Bauer, 2008*; *Suzuki & Hayashi, 2010*). This means that the

m. scapulohumeralis anterior is likely to contribute to propulsion with relatively little muscle force. The m. scapulohumeralis anterior inserts proximally and posterodorsally into the lepidosaur humerus, relatively proximal to the m. latissimus dorsi (*Jenkins & Goslow, 1983*; *Russell & Bauer, 2008*). Thus, we posit that in *C. eurymerus* possibly an insertion of the m. scapulohumeralis anterior proximal and posterodorsal into the humerus was present, rather proximal to the insertion of the m. latissimus dorsi on the dorsal tuberosity. This insertion of the m. scapulohumeralis anterior could be indicated by severe rugosities on the dorsal tuberosity of the humerus.

### Implications of FESA for hindflipper musculature reconstruction

The reconstructions of the femur muscles followed biological principles (*Krahl & Witzel, 2021*). The muscles were tested for functionality on a mechanical basis in the following text. Based on the FESA, some possible muscle origin and insertion areas are functionally favorable over others. An origin of the m. ambiens at the pubic tuberosity is well supported by the EPB. An origin ventrally below the acetabulum would also be well supported by the EPB. From a mechanical point of view, the former has much better leverage than the latter because the pubis and ischium are modified into almost flat ventrally located plates. An origin area of m. ambiens below the acetabulum can therefore be discarded.

The origin of the m. iliofemoralis was reconstructed on the lateral surface of the ilium. *Krahl & Witzel (2021)* also discuss the possibility, weakly supported by the EPB (only in turtles), that the area of origin of the m. iliofemoralis may have spread to the vertebral column (*Zug, 1971*; *Walker, 1973*). An origin on the vertebral column would improve the lever arm of this muscle. When the femur is depressed, the m. iliofemoralis parallels the dorsal trochanter of the femur and wraps around it, exerting compressive stress on it. The ilium is greatly reduced in size, about the same size as the scapula. This would suggest either a greatly reduced m. iliofemoralis or that its site of origin has shifted to the vertebral column. This attachment shift in turn would have removed the size constraint on this muscle attachment surface by the limited attachment area caused by the strong evolutionary size reduction of the ilium in plesiosaurs. Alternatively, the shifting of muscles like the m. iliofemoralis from the ilium onto the vertebral column may have freed the ilium from the constraint to provide a sufficient attachment area for locomotory muscles and could have allowed for its significant size reduction.

According to the EPB, four areas of origin for m. puboischiofemoralis internus are equally possible: a large area of origin on most of the dorsal pubis, a smaller origin on the anterodorsal ischium, a small area of origin on the medial and ventral ilium, and an origin on the vertebral column. Reconstruction of the LOA of the m. puboischiofemoralis internus shows that they wrap around the dorsal trochanter of the plesiosaur femur. This implies that the m. puboischiofemoralis internus contributes significantly to the elevation of the femur. A large pubic ramus and the ischiadic and vertebral portions running parallel to the dorsal trochanter can be justified from a mechanical point of view. An iliac origin seems rather unlikely, as it would wrap around the femoral trochanter and then around the anterior part of the ilium. Since the ilium is reduced, this muscle portion would be small. In addition, no change in muscle length could be measured. All of

this suggests that an iliac origin of the m. puboischiofemoralis internus is unlikely, at least mechanically. An iliac origin has low leverage because it is almost in the same plane as the femur and results in little muscle shortening, whereas an origin from the vertebral column provides a larger lever arm and would certainly have been favorable from a mechanical point of view.

The m. gastrocnemius internus originating from the tibial epicondyle, is equally well supported by the EPB as originating at the distal tibial epicondyle or at the proximal tibia. The FESA clearly shows that an origin similar to that of the m. gastrocnemius externus more proximal to the tibial epicondyle at the point where the femur anterodistally and posterodistally deflects, is preferable. Otherwise, it would be problematic to load the extended femoral epicondyle by compressive stress in the FESA (see below).

## Reconstructed agonists and antagonists in the limbs of plesiosaurs
### Agonists and antagonists of the forelimbs and their function

M. coracobrachialis brevis, m. coracobrachialis longus, m. biceps brachii, the large posterior portions of m. pectoralis, m. subcoracoscapularis, m. supracoracoideus and m. latissimus dorsi are humeral retractors. They are opposing the humeral protractors, namely the m. deltoideus clavicularis, m. deltoideus scapularis, and the small anterior portions of the m. supracoracoideus, m. subcoracoscapularis, and m. latissimus dorsi. The m. deltoideus scapularis, the m. subcoracoscapularis, the m. latissimus dorsi, and possibly to a very small extent the m. triceps brachii, the m. scapulohumeralis anterior, and the m. scapulohumeralis posterior cause elevation of the humerus. The humeral depressors m. supracoracoideus, m. coracobrachialis brevis, m. coracobrachialis longus, m. deltoideus clavicularis, m. biceps brachii, and m. pectoralis act antagonistically to the humeral elevators. The m. triceps brachii, the smaller anterior portion of the m. subcoracoscapularis, the large posterior portion of the m. pectoralis, the m. biceps brachii, the m. scapulohumeralis anterior, the m. scapulohumeralis posterior, and the m. deltoideus clavicularis contribute to the downward rotation of the foreflipper. Functionally antagonistic are m. deltoideus scapularis, m. coracobrachialis brevis, m. coracobrachialis longus, the larger posterior part of m. subcoracoscapularis and m. latissimus dorsi (Table 2).

Further subdivisions into agonistic and antagonistic muscles are possible: The small anterior portion of the m. latissimus dorsi (providing weak elevation and protraction) is opposing in function to the large posterior portion of the m. pectoralis (providing strong retraction and depression). The small anterior m. pectoralis portion (providing protraction and depression) and the large posterior m. latissimus dorsi portion (providing strong elevation and retraction) act as agonist and antagonist. The m. subcoracoscapularis and m. deltoideus scapularis, which elevate and retract the humerus, find their functional antagonists in the m. coracobrachialis brevis, m. coracobrachialis longus, m. biceps brachii, and the posterior portion of the m. supracoracoideus, which depress and retract the humerus. The large posterior portion of the m. subcoracoscapularis acts as a humeral elevator and retractor, and the anterior

portion of the m. supracoracoideus and m. deltoideus clavicularis act opposite to them as humeral depressors and protractors (Table 2).

Muscles originating from the humerus aid in flipper twisting (*Krahl & Witzel, 2021*): the m. flexor carpi ulnaris could have displaced the ulnar side of the carpus ventrally relative to the humerus (or flex metacarpal V), while the m. extensor carpi ulnaris and humeral m. triceps brachii could have displaced the ulna in the opposite direction, and the former could have possibly extended metacarpal V. The m. flexor carpi radialis could either have flexed metacarpal I or displaced the radial side of the carpus ventrally. The m. brachialis and m. pronator teres also contribute to the latter function. These are antagonistically opposed by m. supinator longus + m. extensor carpi radialis. The m. flexor digitorum longus (flexion of digits I to V) is opposed by the m. extensor digitorum communis and the digital extensors (Table 2).

## Hindlimb agonists and antagonists and their function

The elevators of the hindflipper are m. puboischiofemoralis internus, m. iliotibialis, m. iliofemoralis, m. iliofibularis, m. caudofemoralis brevis, m. caudofemoralis longus, m. flexor tibialis externus (iliac portion), and m. flexor tibialis internus (portion originating from the vertebral column). M. puboischiofemoralis externus, m. adductor femoris, m. ischiotrochantericus, m. puboischiotibialis, m. flexor tibialis externus (ischial portion) and m. flexor tibialis internus (ischial portion) act as femoral depressors. Protractors are the m. puboischiofemoralis externus (pubic portion), m. puboischiofemoralis internus (pubic portion and vertebral column portion), m. ambiens, and m. pubotibialis. The retractors of the hindflipper are m. puboischiofemoralis externus (ischial portion), m. puboischiofemoralis internus (ischial and iliac portions), m. adductor femoris, m. ischiotrochantericus, m. iliofemoralis, m. iliotibialis, m. caudofemoralis brevis, m. caudofemoralis longus, m. flexor tibialis externus, and m. flexor tibialis internus. The downward rotation of the flipper leading edge during the downstroke is controlled by the agonists m. puboischiofemoralis internus (pubic portion), m. puboischiofemoralis externus (ischial portion), m. caudofemoralis brevis, m. caudofemoralis longus, m. ambiens (when the femur is elevated), m. ischiotrochantericus, m. iliofibularis (as long as the fibula is below the origin of these muscles), m. puboischiotibialis, m. pubotibialis (when the femur is elevated), m. flexor tibialis internus and m. flexor tibialis externus. The agonists are opposed by the antagonistically acting m. iliotibialis, m. ambiens (when the femur is depressed), m. pubotibialis (when the femur is depressed), m. iliofemoralis, m. puboischiofemoralis internus (ischial and iliac portion) and m. puboischiofemoralis externus (large pubic portion) (Table 3).

In addition, the muscles can be assigned to the following subgroups: The m. puboischiofemoralis internus, which originates from the dorsal pubis and vertebral column, is opposed by muscles originating from the ventral ischium (m. puboischiofemoralis externus (ischial portion), m. adductor femoris, and the ischial portions of m. flexor tibialis internus and m. flexor tibialis externus). The pubic portion of m. puboischiofemoralis externus, m. ambiens and m. pubotibialis, which originate

from the ventral region of the pubic bone, have muscles originating from the ilium and posterior vertebral column as antagonists (m. flexor tibialis internus and m. flexor tibialis externus, m. iliofibularis, m. iliotibialis, m. iliofemoralis, m. puboischiofemoralis internus (ischial portion)). Musculus ambiens and m. pubotibialis have m. iliofibularis as antagonist. M. femorotibialis and m. extensor digitorum longus mostly originate dorsal to the femur. M. gastrocnemius externus, m. gastrocnemius internus and m. flexor digitorum longus arise ventrally from the femur. M. gastrocnemius externus and m. gastrocnemius internus as well as m. flexor digitorum longus (flexion of the toes) appear to be opposite to m. extensor digitorum longus (extension of the toes) and m. femorotibialis (Table 3).

## Muscle forces and length changes

### Muscle forces and FESA

#### Humeral musculature

Compressive stress distribution and muscle forces were calculated for the downstroke and upstroke load cases of the humerus FE model. For the initial FESA runs, the dorsal tuberosity of the humerus was left unloaded. Augmenting the FESA to include wrapping of the m. latissimus dorsi and m. subcoracoscapularis around the dorsal tuberosity, which resulted from LOA observations, helped to apply compressive stress to this process. In addition, extensor and flexor muscles wrapping around the anterior and posterior distal processes (ectepicondylar and entepicondylar processes) of the humerus allowed their loading by compressive stress. This is because muscles that wrap around a bone exert compressive stress on it. In contrast, muscles that do not wrap around the bone exert only local tensile stress on the bone.

In the FESA results for the humerus (Figs. 9C, 9D, 9G, 9H, 9K and 9L), the colors red, orange, and yellow correspond to low compressive stress (0 to −3.6 MPa). The dorsal tuberosity of the humerus, as well as large portions of its distal region, are subjected to low compressive stress. This agrees well with the observation that the distal humerus consists largely of spongy bone covered only by a thin layer of cortical bone (cf. *Krahl & Witzel, 2021*). The green color spectrum correlates with moderate compressive stresses (−3.6 to −7.2 MPa). Large areas of cortical and cancellous bone, especially in the humeral shaft, but also smaller areas of the distal region are moderately compressed (Figs. 9C, 9D, 9G, 9H, 9K and 9L). Turquoise to blue colors correspond to high compressive stress (−7.2 to −10.8 MPa). The high compressive stress corresponds in part to the outermost cortical layer. In particular, the proximal region of the head and the proximal shaft are subjected to high compressive stress (Figs. 9C, 9D, 9G, 9H, 9K and 9L). The shear stresses for both superimposed load cases were 2.81 MPa (SD 5.4 MPa), well below the shear strength of bone of 62–72 MPa (*Reilly & Burstein, 1975*).

On the distal articular surface of the humerus, high stress peaks occur in a very confined manner (Figs. 9C, 9G and 9K). These are artifacts due to the application of the counterforce. The counterforce was applied scattered over the large distal articular surface instead of applying it to one point to obtain a more realistic force induction.

However, this is a compromise between realistic conditions and the technical capabilities of the software.

On the downstroke, the m. pectoralis develops the highest muscle force (9,600 N) among the muscles spanning the glenoid. Surprisingly, on the upstroke, the m. pectoralis still develops a higher force (5,267 N) than either of the two main humeral elevators, *i.e.*, m. latissimus dorsi and m. subcoracoscapularis. Nevertheless, both, m. subcoracoscapularis (4,422 N) and m. latissimus dorsi (3,918 N), develop high forces to maintain the upstroke together. In general, it appears that greater muscle forces are generated by the retractors and depressors of the humerus than by its elevators and protractors. In addition, extensors and flexors sometimes develop extremely high muscle forces, with the m. flexor carpi radialis generating 8,460 N on the downstroke and the m. extensor digitorum communis generating 6,000 N on the upstroke (Table 4).

There is a wide range of values for the Young's modulus of bone in the literature, and the value of 12,000 MPa chosen in this study is a relatively low estimate. We justify our choice of this low value because the flipper model is made of a homogeneous material. A higher value would not change the stress distribution or the stress level but would affect the strain: a higher Young's modulus is associated with lower strains in linear elastic materials.

## Femur muscles

The compressive stress distribution and muscle forces were calculated for the down- and upstroke load cases of the femur FE model. In the initial femur FESA runs, we were unable to apply compressive stresses to the dorsal trochanter and distal epiphyses of the femur because the muscles would simply pull away from their origin. Thus, only localized tensile stresses were observed in the FESAs. We then introduced muscle wrappings as in the humerus, with the m. iliofemoralis and m. puboischiofemoralis internus wrapped around the dorsal trochanter and the extensors and flexors wrapped around the distally greatly expanded epicondyles of the femur. This resulted in these structures being loaded by compressive stress.

Color coding of compressive stress distribution in the plesiosaur femur (Figs. 10C, 10D, 10G, 10H, 10K and 10L) is the same as for the humerus (see above). Low compressive stress correlates mainly with the medullary region in the mid to distal femur and the distal extensions of the femur. Moderate compressive stresses occur mainly in regions where cortical bone is present, especially on the outer femoral shaft. High compressive stress values correspond mostly to regions of the femoral head and part of the cortical bone of the proximal shaft (Figs. 10C, 10D, 10G, 10H, 10K and 10L). As in the FESA of the humerus, the localized compressive stress peaks on the distal articular surface of the plesiosaur femur were due to the selective application of the counterforce at several points scattered across the articular surface (see above) (Figs. 10C, 10G and 10K). The shear stresses were 2.56 MPa (SD 2.21 MPa).

The muscle forces of the many two-joint muscles (m. pubotibialis, m. puboischiotibialis, m. flexor tibialis externus, m. flexor tibialis internus, m. ambiens, m. iliotibialis, m. iliofibularis) in the hindflipper cannot be determined, because they influence the femur

**Table 5 Muscle forces of *Cryptoclidus eurymerus* (IGPB R 324) femur by superposition of FESA load cases.**

| Muscle | Muscle force (N) |
| --- | --- |
| m. puboischiofemoralis externus | 7,878 |
| m. puboischiofemoralis internus | 7,611 |
| m. femorotibialis | 1,521 |
| m. adductor femoris | 3,938 |
| m. ischiotrochantericus | 984 |
| m. iliofemoralis | 253 |
| m. caudifemoralis brevis | 506 |
| m. caudifemoralis longus | 507 |
| m. extensor digitorum communis | 1,014 |
| m. gastrocnemius | 1,176 |
| m. flexor digitorum longus | 786 |

only indirectly by contributing to the counterforce. On the downstroke, the m. puboischiofemoralis externus generates the highest muscle force (7,878 N). On the upstroke, the m. puboischiofemoralis internus generates up to 7,611 N. The forces of the extensor and flexor muscles are much lower in the femur than in the humerus. The gastrocnemius muscle, a flexor muscle, develops a total force of up to 1,176 N (Table 5).

## Changes in length of the limb muscles

Muscles originating dorsally from the glenoid and acetabulum extend on the downstroke and contract on the upstroke. Muscles originating ventrally from the glenoid and acetabulum contract when the humerus and femur are depressed on the downstroke and lengthen on the upstroke when the humerus and femur are elevated.

The total muscle length changes of the foreflipper vary from 0% to 70.87%. Three muscles (or parts of them) (m. deltoideus clavicularis, m. triceps brachii (anterior and posterior part)) show no length change, *i.e.* the length changes were not measurable with the technique used here, *i.e.* they are smaller than 1.7 cm (width of the terminal strips). The m. coracobrachialis brevis (posterior part) shows very small muscle shortening (3.88%). Otherwise, the changes in total muscle length cover the entire physiological spectrum, from about 9% in the posterior part of the m. subcoracoscapularis to 37% in the anterior part of the m. latissimus dorsi. The only muscle that stands out is the m. deltoideus scapularis with a total length change of over 70%. This is clearly not physiological. Therefore, a screw eye pin was screwed into a different hypothetical region of origin of the m. deltoideus scapularis. This different origin was located at the ventral to ventrolateral scapula anterior to the glenoid. The total muscle length change was again measured in all three flipper positions and was now within the measurement error of the analog model and well within physiological limits (Table 1).

Total muscle length changes for the hindflipper range from 0% to 35.8%. We detected no length changes for the m. caudofemoralis brevis (ilium portion) and the m. pubotibialis.

In addition, the m. caudofemoralis brevis, which originates from the vertebral column, shows little change in total muscle length (5.2%), while the muscle with the greatest change (35.82%) in total length is the part of the m. puboischiofemoralis internus that originates from the vertebral column (Table 1).

Considering the agonistic and antagonistic muscles, the total length changes of m. pectoralis and m. latissimus dorsi, the two muscles that mainly drive the downstroke and upstroke of the foreflipper, are quite similar: the anterior part of m. latissimus dorsi (36.96%) and the posterior part of m. pectoralis (35.65%), and the posterior part of m. latissimus dorsi (21.92%) and the anterior part of m. pectoralis (18.75%). We expected that the overall changes in muscle lengths of agonists and antagonists would be similar because of their opposite functions. Instead, we found that the muscles exhibiting comparable total length changes were more likely to be determined by their geometric arrangement relative to the glenoid or acetabulum. That is, a muscle originating from the posteroventral ischium (*e.g.*, m. puboischiofemoralis externus (23.55%)), for example, exhibited similar shortening to a muscle originating from the anterodorsal pubis (m. puboischiofemoralis internus (21.04%)).

## DISCUSSION

### Myology

The muscle reconstructions on which this study is based were obtained by evaluating comparative anatomical data (*i.e.*, using the EPB) (*Krahl & Witzel, 2021*). These biologically derived muscle reconstructions are examined to determine whether they also meet the mechanical criteria to which muscles are subjected. In most cases, we find that the reconstructed muscles also meet the mechanical criteria. However, some fore- and hindflipper muscle reconstructions (m. biceps brachii, m. deltoideus scapularis, m. gastrocnemius internus) were modified for mechanical reasons to obtain a more homogeneous compressive stress distribution in the FESA of the humerus and femur. We found mechanical evidence to support some reconstructions of *Krahl & Witzel (2021)* that were rather weakly supported by the EPB (*i.e.*, coracoid portion of the m. subcoracoscapularis, m. coracobrachialis brevis, m. coracobrachialis longus, m. ambiens, m. iliofemoralis, m. puboischiofemoralis internus and externus).

One muscle, the m. scapulohumeralis anterior, was added here to the reconstructions of *Krahl & Witzel (2021)*, as it may aid in rotation of the humerus. We reconstruct the area of origin of the m. scapulohumeralis anterior on the anterior scapular blade. Compared to the reconstruction of *Araújo & Correia (2015)*, this muscle is located less ventrally and more laterally in our reconstruction. *Robinson (1975)* and *Watson (1924)* reconstructed its attachment surface on the medial and ventral regions of the scapula, which is not fully supported by the EPB (*Jenkins & Goslow, 1983*; *Russell & Bauer, 2008*). In contrast, *Carpenter et al. (2010)* reconstructed a large area of origin of the m. scapulohumeralis anterior on the lateral aspect of the scapula. Instead, we reconstructed a large m. deltoideus scapularis in approximately the same area, which is better supported by the EPB (*Walker, 1973*; *Meers, 2003*; *Russell & Bauer, 2008*; *Suzuki & Hayashi, 2010*). The results presented here disagree with *Tarlo (1958)* in that the origin of

the m. scapulohumeralis anterior is on the anteroventral part of the scapula, as this is not supported by the living Sauropsida (*Walker, 1973*; *Meers, 2003*; *Russell & Bauer, 2008*; *Suzuki & Hayashi, 2010*). The muscle reconstructions by *Lingham-Soliar (2000)* are merely schematic. It is impossible to determine the exact muscle attachments from his drawings. However, judging from their geometric arrangement, his muscle reconstructions are similar to our results.

M. scapulohumeralis anterior occurs exclusively in lepidosaurs and not in turtles or crocodylians (*Walker, 1973*; *Meers, 2003*; *Russell & Bauer, 2008*; *Suzuki & Hayashi, 2010*). In lepidosaurs, it inserts posterodorsally into the humerus; therefore, according to the EPB, this pattern was transferred to the plesiosaur in this study. None of the previous authors who reconstructed this muscle reconstructed its insertion at this location. They placed it either anterodorsally on the humerus (*Watson, 1924*; *Tarlo, 1958*; *Robinson, 1975*) or on the dorsal surface of the humerus (*Lingham-Soliar, 2000*; *Carpenter et al., 2010*).

We reconstructed m. scapulohumeralis anterior in agreement with *Tarlo (1958)* and *Watson (1924)* as possibly subordinately elevating and rotating the humerus (*Watson, 1924*; *Tarlo, 1958*). *Watson (1924)* and *Tarlo (1958)* disagree on how the extension and rotation occurred, whereas *Watson (1924)* suggested that the muscle rotated the humerus anteriorly upward, *Tarlo (1958)* concluded the opposite. Thus, *Tarlo (1958)*, *Robinson (1975)* and the present study agree on the direction of humeral rotation. However, *Robinson (1975)* additionally reconstructed the m. scapulohumeralis anterior as a depressor, which is contrary to the results of this study. The possible minor elevation function has not been described by any previous author.

## Physiology of the muscles
### Changes in total muscle length
Total length changes were calculated for glenoid and acetabular muscles and tested to see if they were within physiological limits. If it had originated from the lateral scapula, the total length change of the m. deltoideus scapularis is not physiological and would not allow the muscle to produce much force (see *Biewener & Roberts, 2000*) (Table 1; Fig. 8A). This may suggest that the reconstruction of the origin of the m. deltoideus scapularis on the scapula by *Krahl & Witzel (2021)* is incorrect, although well supported by the EPB, and that its origin is restricted to the ventral side of the scapula. An origin of the m. deltoideus scapularis on the ventral to ventrolateral region of the scapula anterior to the glenoid would solve this problem and result in a length change that is within physiological limits (Table 1). Furthermore, a length change within the measurement error, *i.e.*, a small change, would better account for the often non-parallel and rather complex architecture of the deltoid muscle (see, *e.g.*, *Krahl et al., 2019*; *Walker, 1973*; *Meers, 2003*; *Russell & Bauer, 2008*). Removing the m. deltoideus scapularis from the scapular blade would mean that no locomotor muscles attached to the lateral scapular blade in plesiosaurs. Only muscle attachments that suspend the shoulder girdle would remain on the scapular blade. In addition to aquatic adaptation (*Krahl et al., 2019*), this could be another explanation for why the osseous dorsal scapular process is so much smaller than in living Sauropsida (cf. *Walker, 1973*; *Meers, 2003*; *Russell & Bauer, 2008*; *Suzuki & Hayashi, 2010*) and

possible functional analogs (cf. *Walker, 1973*; *English, 1977*; *Schreiweis, 1982*; *Louw, 1992*; *Cooper et al., 2007*). If there was suprascapular cartilage in plesiosaurs, it was probably not very large (*Carpenter et al., 2010*). Similarly, displacement or reduction in size of muscles originating from the ilium (*e.g.*, m. iliofemoralis) could free or largely free the ilium from locomotor muscles, allowing for its reduction in size.

Several muscles (*e.g.*, m. deltoideus clavicularis, both m. triceps brachii heads, m. coracobrachialis brevis (posterior part); m. caudofemoralis brevis (ilium), m. pubotibialis) have similar length changes for elevating and depressing the humerus and femur. If protraction and retraction were also considered, as well as rotation of the humerus about the long axis, the muscle length changes would give different results for all these muscles. It is possible that these nearly isometrically contracting muscles had a complex muscle architecture (see *Biewener & Roberts, 2000* for a review). In addition, it is possible that the m. triceps brachii and the m. pubotibialis had long tendons (*i.e.*, a large noncontractile component). Another possibility is that the muscles that do not show length changes were reduced in plesiosaurs. Some clues might be provided by the EPB: The m. triceps brachii is greatly reduced or completely reduced in Chelonioidea, depending on the species. The m. coracobrachialis brevis is significantly reduced in Testudines in general (*Walker, 1973*). It is possible that the origin of the m. caudofemoralis brevis from the ilium was reduced in plesiosaurs and that the m. caudofemoralis brevis arose only from the vertebral column. The m. pubotibialis is absent in crocodylians and may have been reduced in plesiosaurs as well (*Otero, Gallina & Herrera, 2010*; *Suzuki et al., 2011*).

The muscle length changes of the agonistic and antagonistic muscles of the pelvic and pectoral girdles do not agree well. This could be due to differences in the morphology and geometry of the pectoral and pelvic girdles. On the other hand, the results could also change if flipper protraction and retraction were considered, since the current results focus on the main direction of movement of the flippers: elevation and depression.

Furthermore, muscle architecture and tendon length have not been derived for extinct tetrapods and probably cannot be. The analog approach presented here has the advantage of providing intuitively accessible results. However, using multibody dynamics, sensitivity analyses could be performed much more quickly, results would be more easily reproducible, and coupling multibody dynamics and FEA would allow the interaction of motion and internal stresses to be better studied. The creation of a multibody simulation of the flipper should therefore be a goal of future research.

## Muscle forces
### Muscle redundancy

The analyses presented here show that muscle forces lead to a homogeneous stress distribution of the humerus and femur in the plesiosaur. However, because the flipper models both consist of more muscles than degrees of freedom, these systems are hyperstatic. The redundancy of muscles leads to a theoretically infinite number of muscle activation patterns (*Bernstein, 1967*). Therefore, the muscle forces represent only one possible solution, and it is likely that other muscle forces could be found that would lead to

a homogeneous stress distribution. On the other hand, it was not possible with the optimization algorithm to set the muscle forces to zero, limiting the number of possible solutions (*Kutch & Valero-Cuevas, 2011*). Nevertheless, our results show that the muscle forces required to implement the up- and downstroke can lead to a minimization of the bending of the humerus and femur.

## Musculature of the humerus

On average, the forces of the muscles involved in the downstroke, *i.e.*, by the humeral and femoral depressors and retractors, appear to be higher than those of the muscles involved in the upstroke (Tables 4 and 5). This could mean that in plesiosaurs the downstroke of the fore- and hindflippers was stronger than the upstroke. Similarly, the downstroke of the foreflipper is more powerful than the upstroke in Cheloniidae (*Davenport, Munks & Oxford, 1984*; *Krahl et al., 2019*). However, this difference in efficiency is not found in all underwater fliers, as it does not occur in penguins, where the downstroke of the wing is as efficient as the upstroke (*Clark & Bemis, 1979*).

As reconstructed here, the m. scapulohumeralis anterior and posterior appear to be exclusively humeral rotators, as they operate with considerably high muscle forces during the downward stroke (Table 4). In this way, their lifting function is low. This is surprising because both muscles tend to be small in living lepidosaurs (*Jenkins & Goslow, 1983*; *Russell & Bauer, 2008*). Either this means that their areas of origin and attachment were enlarged in plesiosaurs, or the muscle forces of m. scapulohumeralis anterior and posterior would need to be tested with further FESA runs. These would show whether a similar homogeneous compressive stress distribution can be obtained by redistributing a large portion of the force to muscles with similar function and LOA, *e.g.*, m. latissimus dorsi, m. subcoracoscapularis, *etc*. The LOA of m. scapulohumeralis anterior and m. scapulohumeralis posterior, which would wrap around the dorsal tuberosity in an unusual manner from posterior to anterior, seem to support the hypothesized redistribution of force to another muscle as described above.

A surprising result is that the m. pectoralis is the muscle that also develops the highest muscle force during the elevation of the foreflipper (Table 4). Overall, it should be assumed that humeral elevators and protractors should be capable of providing higher power than depressors and retractors, otherwise foreflipper elevation and protraction becomes impossible. In addition, the pectoral girdle is suspended by muscles and tendons from the vertebral column, thorax, and gastralia (*e.g.*, (*Avery & Tanner, 1964*; *Walker, 1973*). A swinging pectoral and pelvic girdle may contribute significantly to locomotion in living Otariinae, Testudines, and Crocodylia (*Walker, 1971a*; *Baier & Gatesy, 2013*; *Mayerl, Brainerd & Blob, 2016*; *Schmidt, Mehlhorn & Fischer, 2016*). For a long time, the importance of the contribution of pectoral and pelvic girdle swinging to locomotion in Tetrapoda has been neglected and underestimated, and it certainly requires further research. In extinct marine reptiles, a swinging pectoral and pelvic girdle has never been considered, as far as the authors are aware. An actively swinging pectoral and pelvic girdle could contribute to the range of motion of the plesiosaurs' fore- and hindflippers and the total force with which the flippers were moved. Especially in the shoulder region of

plesiosaurs, where there is no bony or cartilaginous connection to the trunk (unlike to the pelvic region), strong shoulder muscles and ligaments connecting the pectoral girdle to the vertebral column, ribs, and gastralia would be necessary.

Acting on the region of the dorsal the foreflipper, the m. extensor carpi ulnaris develops rather low muscle forces (1,000 N) compared to the m. extensor digitorum communis (6,000 N). M. flexor carpi radialis and m. flexor digitorum longus are topologically arranged similarly to m. extensor carpi ulnaris and m. extensor digitorum communis, but on the ventral side of the foreflipper. In contrast, the m. flexor carpi radialis and m. flexor digitorum longus develop much lower muscle forces (both 1,500 N) (Table 4). However, extensor and flexor forces were calculated based only on their contribution to physiological loading of the femur epicondyles. Extending the model to include the rest of the flipper bones could help define them more precisely.

## Musculature of the femur

When comparing the muscle forces of the extensor and flexor muscles of the plesiosaur foreflipper and hindflipper, it is noticeable that those acting on the femur (Table 5) are generally lower than those acting on the humerus (Table 4). This is because there are many more two-joint muscles and fewer, as well as less independently working, extensor and flexor muscles in the plesiosaur hindflipper than in the foreflipper. The two-joint muscles also aid in femur protraction/retraction, elevation/depression, and knee flexion in present-day Sauropsida (*Snyder, 1954*; *Walker, 1973*; *Otero, Gallina & Herrera, 2010*; *Anzai et al., 2014*). Because the plesiosaur knee was immobile, these muscles were interpreted by *Krahl & Witzel (2021)* as part of the flipper longitudinal axis twisting mechanism in addition to their functions as depressors/elevators and protractors/retractors. Therefore, it is possible that the numerous two-joint muscles of the plesiosaur hindflipper partially supported functions that were taken over by the much more differentiated extensor and flexor muscles of the front flipper (*Krahl & Witzel, 2021*). The forces of the two-joint muscles could not be determined in this study, as they only indirectly influence the FESA by increasing the counterforce imposed by the tibia and fibula. Therefore, it is expected that the single extensor and the two flexors of the hindflipper would have lower overall muscle forces than the numerous extensors and flexors of the front flipper. Finally, the femur and humerus differ morphologically in their distal region in *Cryptoclidus* (as opposed to many other plesiosaurs), and it is possible that the hindflipper contributes less to propulsion than the foreflipper, as suggested by *Lingham-Soliar (2000)* and *Liu et al. (2015)* for plesiosaurs in general.

## Comparison with the cheloniid humerus

There are some similarities in the muscle forces of certain muscles in *Cryptoclidus eurymerus* and in Cheloniidae: The m. pectoralis develops the highest force of all muscles that attach proximally to the humerus (*Krahl et al., 2019*). In addition, of the major humeral elevators, the m. subcoracoscapularis produces higher forces than the m. latissimus dorsi in both taxa (*Krahl et al., 2019*). M. coracobrachialis brevis develops less force in sea turtles (*Krahl et al., 2019*) than in the plesiosaur. In contrast, the

m. coracobrachialis longus develops higher forces in sea turtles (*Krahl et al., 2019*) than in the plesiosaur. While m. deltoideus scapularis and m. deltoideus clavicularis contribute to propulsion in very different ways in Cheloniidae (*Krahl et al., 2019*), they act with broadly similar forces in the plesiosaur.

Another difference between the forces of the upper arm muscles of cheloniids (*Krahl et al., 2019*) and plesiosaurs is that the forces vary by an order of magnitude in the former, whereas they do not necessarily differ greatly in the plesiosaur. These results support a hydrodynamic study of *Cryptoclidus eurymerus* showing that pectoral flipper twisting is critical for underwater flight in plesiosaurs (*Witzel, Krahl & Sander, 2015*; *Witzel, 2020*). Moreover, these results confirm the myological mechanism of flipper twisting proposed by *Krahl & Witzel (2021)*. In general, the higher muscle forces in the plesiosaur could be due to scaling effects, *i.e.*, the generally larger relative body size of plesiosaurs compared to sea turtles.

The muscle bellies generating these enormous muscle forces (up to 9,600 N) in the shoulder girdle were large and occupied a lot of space. Therefore, it is problematic that the extensor and flexor muscles were inferred to generate similarly large forces because their potential areas of origin are much smaller than those of the glenoid-spanning muscles. There may be several solutions to this paradox. For example, the m. flexor digitorum longus in Sauropsida has a second head originating from the carpus (*Walker, 1973*; *Meers, 2003*; *Russell & Bauer, 2008*), so it may have been able to develop a much greater force than just through the humeral muscle belly. In addition, these muscles may have a complex architecture that saves space compared to the muscles of the pectoral girdle. Long tendon structures may have been a mechanism for conserving energy during locomotion (*Roberts et al., 1997*; *Biewener & Roberts, 2000*).

Another possibility, how large muscle forces in distal muscles may have been attained, can be observed in dolphins. Dolphins have a relatively well ossified flipper skeleton, although essentially no individual muscles are recognizable. They have only layers of parallel, fibrous connective tissue covering the flipper bones (see *Cooper et al., 2007*, Fig. 4, p. 1128). Conversely, this means that the hydrodynamic forces plus the "muscle" force that these connective tissue layers can exert are, in sum, sufficient to cause ossification of the flipper bones. Similar aponeurotic stratification, perhaps also directed in the main directions of flipper twisting, could account for a significant portion of the muscle forces calculated with FESA. In addition, the connective tissue covering the broad space of the non-functional elbow joint, carpus, and wrist could passively store energy to compensate for some of the calculated forces.

## SUMMARY AND CONCLUSIONS

The highly aquatic-adapted locomotory system of plesiosaurs changed little throughout the 135 million years of plesiosaur evolution. The question of whether plesiosaurs rowed, flew underwater, or used a combination of both has not yet been fully answered. Here we present analog muscle reconstructions and a computer model of the plesiosaur *Cryptoclidus* based on the mounted skeleton IGPB R 324. These models are consistent with

underwater flight in plesiosaurs based on comparative anatomical and muscle physiological data in accordance with mechanical principles.

We show that it is possible to test muscle reconstructions and associated lines of action with FESA by targeting a homogeneous compressive stress distribution in the humerus and femur. The muscle reconstructions of *Krahl & Witzel (2021)* were largely confirmed but were also supplemented and corrected with FESA. As in Cheloniidae, muscles that wrap around bony processes, *i.e.*, the dorsal tuberosity of the plesiosaur humerus and the dorsal trochanter of the plesiosaur femur, as well as their epicondyles, proved to be necessary to load the bony structures. Measurements in an analog model of the total length changes of all muscles attaching to, originating from, and spanning the humerus and femur of a plesiosaur revealed that an origin of the m. deltoideus scapularis at the lateral scapula is unphysiological. This suggests a reduction of this muscle from this part of its site of origin, which is well documented in Sauropsida (*Walker, 1973*; *Meers, 2003*; *Russell & Bauer, 2008*). Muscle forces show some agreement with cheloniid humerus muscles, but also differences, underscoring that underwater flight was achieved in a convergent manner in both lineages. High extensor and flexor muscle forces in plesiosaurs confirm the hypothesis that twisting around the flipper long axis was essential for underwater flight in plesiosaurs (see chapter above (Comparison to Cheloniidae humerus)).

## ACKNOWLEDGEMENTS

We thank Aart Walen (Creatures and Features), Arnhem, The Netherlands, for kindly providing casts of the pectoral and pelvic girdles and the fore- and hindflippers of IGPB R 324, which were crucial for the analog model used in this study. We thank L. Baumeister of the Biomechanics Research Group, Chair of Product Development, Faculty of Mechanical Engineering, Ruhr University Bochum, Germany, for providing FE models of the femur. We thank Tanja Wintrich (Bonn, Germany) for making the micro-CT scans of the humerus and femur of IGPB R 324. Finally, we thank the three reviewers, Colin Palmer, Stephan Lautenschlager, and Eric Snively, and the editor Mark Young for their constructive criticism, which resulted in a greatly improved manuscript.

### Funding

This study was funded by Deutsche Forschungsgemeinschaft (DFG) grant WI1389/8-1. The funders had no role in study design, data collection and analysis, decision to publish, or preparation of the manuscript.

### Grant Disclosures

The following grant information was disclosed by the authors:
Deutsche Forschungsgemeinschaft (DFG): WI1389/8-1.
## Competing Interests

The authors declare that they have no competing interests.

## Author Contributions

- Anna Krahl conceived and designed the experiments, performed the experiments, analyzed the data, prepared figures and/or tables, authored or reviewed drafts of the paper, and approved the final draft.
- Andreas Lipphaus performed the experiments, prepared figures and/or tables, authored or reviewed drafts of the paper, and approved the final draft.
- P. Martin Sander conceived and designed the experiments, authored or reviewed drafts of the paper, and approved the final draft.
- Ulrich Witzel conceived and designed the experiments, performed the experiments, authored or reviewed drafts of the paper, and approved the final draft.

## Data Availability

The humerus and femur models are available in the Supplemental Files. They are the basis for the finite element analyses.

## Supplemental Information

Supplemental information for this article can be found online at http://dx.doi.org/10.7717/peerj.13342#supplemental-information.

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
