# Peer review of "Determination of muscle strength and function in plesiosaur limbs: finite element structural analyses of Cryptoclidus eurymerus humerus and femur"

_PeerJ, doi:10.7717/peerj.13342_

## Round 0.1 · original submission · Major Revisions

Dear authors,

Based on the unanimous recommendations of the three reviewers, I am making a 'major revisions' decision.

All three reviewers have provided positive and constructive feedback, which I am sure will strengthen your manuscript.

My own comments: the title needs re-phrasing. Perhaps something like: "Plesiosaur muscle forces and flipper twisting: a finite element structure analysis approach". At the moment it is grammatically clunky. I think you could make it much more succinct and appealing to potential readers.

I agree with reviewer one that the omission of Muscatt et al. is surprising. I also agree with reviewers two and three that your figures will need more work prior to resubmission.

I look forward to receiving your revised manuscript.

·

Basic reporting

See my overall review comments.

Experimental design

See my overall review comments.

Validity of the findings

See my overall review comments.

Additional comments

Review of plesiosaur paper by Krahl et al.

As noted in my email to the editors, I am afraid that there are very large sections in the paper on which I am unqualified to comment. I therefore restrict my review to a few general questions, mainly related to the biomechanics/FEA aspects of the paper.

Overall this appears to be an impressive piece of work, but unfortunately somewhat inaccessible to someone with my background, at least in the context of the time available to undertake a review.

I list them below, line by line.

LL123-125. I am surprised that you do not include Muscutt et al in the this list of references. The paper clearly discusses the different possible propulsion modes.

L139 - flapped would be a better word then beaten.

L160 - I think you could add a bit more here about the way that the correct motion of the hind flippers relative to the vortices shed by the fore-flippers can actually enhance performance.

L199 Fluctuatingly is not the best word here, indeed I am not suer it is in the English dictionary.

LL200 et seq. The sheer density of references here obscure meaning and make it difficult for the reader. Do you really need so many?

L203 - I don’t understand the use of the word subordinately in this context. I have not reviewed all the references given, but my reading of Figure 3 in Biewener & Dial is that the highest strains are in compression, but the authors say that “Shear strains produced by torsion averaged 1.5 times the longitudinal components of strain…”. The values of the strains are only really meaningful if related to the ultimate strength of the material, and it is well established that the tensile strength of bone is not much more than half the value in compression. Torsional strength is also much lower than the compressive value. Thus in pigeons at least, the torsional strains are the critical ones in terms of closeness to material failure.
I therefore do not understand why the authors have focussed on compressive strains.

L210 I do not understand the phrase “By reducing the bending moment, biological lightweight structures may evolve.” More explanation of the mechanism is needed please.

L230-233 The references here are In german, so inaccessible to me. However, I do think the authors would have no trouble in finding papers in the wider aerodynamics/fluid dynamics literature on flapping foil propulsion that show clearly that efficient flapping propulsion requires the foil to move in both heave and pitch. This has been know for many years and is not at all novel.

L240 et seq. The approach of using a physical model is very gratifying to someone of my generation, but was it really the right approach in these days of computer simulation packages such as OpenSim, MDA models such as GaitSym? I think that a few words about this should be included. They have the great advantage that it is much easier to run sensitivity analyses at the tough of a mouse, rather than have to spend ages fiddling with bits of string. It is also well know that the thickness of cartilage has a big effect on joint mobility, so more discussion about what appears to have been some what arbitrary use of styrofoam would be beneficial.

L339 et seq. The text refers to the bone tissue being segmented out, but Figure 7 and the STL files provided only appear to describe the surface of the bone, not the internal structure. What data was used for the internal structure - wall thickness etc? In the absence of this information, it is hard to comment on the validity of the FEA analysis.

L355 In contrast to other statements that are supported by an overabundance of references, the value for E used in the FEA is simply stated with no justification or discussion. The value of E is fundamental to the results of and FEA analysis, so needs to be selected with care. A value of 12GPa could be considered low (see Currey’s book Bones for exampleTable 4.3 wherein many values are in excess of 20GPa.)

L384 - another word I do not recognise: angulated. Angled?

L763 et seq. I think that power and efficiency are being confused here. A very powerful downstroke may produce a lot of thrust, but if the flow regime is non-optimal (eg Strouhal number) than the efficiency may be low. This is unequivocally the case, for example, when birds are taking off.

L879 - why is a homogenous compressive stress distribution chosen as the defining variable. What about tensile or shear, both orientations of lesser ultimate strength in bone?

L891 I have not been able to follow all the detailed anatomical discussions in the paper, so cannot understand the very last statement. At the very least, perhaps the text could refer the reader to the relevant part of the mass of detail?

On the basis of my comments with regards L339, 355, 879 and 891, I have to admit to not being able to form a view on the validity of the findings. Clearly a huge amount of work has gone into the study, so it may be fine, but I think it needs to be explained better for the non-specialist reader.

·

Basic reporting

The submitted manuscript describes the biomechanical study of the fore- and hindflippers of the plesiosaur Cryptoclidus in a mostly clear and professional way. Sufficient background and context is provided in the introduction and referred to in the discussion section as well. The overall structure is logical and most of the individual sections are sufficiently detailed.
I have made some smaller edit and comments (mostly regarding language and clarity) directly in the attached PDF.

Experimental design

The research question is well-defined and clear. The experimental design is described in detail for the physical setup to calculate muscle orientation and attachments. However, the setup of the FEAs is very short and a lot of the relevant steps and underlying assumptions have been omitted.

While the analogue model setup and the approach to obtain muscle lines and attachment angles is clearly described, no details are given on how muscle forces were obtained from the FEAs. "Forces of each humerus and femur muscle were approximated stochastically" and this seems to follow the approach outlined in Sverdlova and Witzel, 2010). In the referred study, in-vivo measurements were used to guide the initial muscle setup. Based on this initial analysis the loading regimes were refined to obtain a tension-free condition. If the same approach was used here, which initial force values or expected reaction forces were assumed and based on which assumptions (approximate muscle cross-section area, body size, etc.)? Further, was the iterative refinement done manually or automated? In both cases, what were the criteria used to guide the refinement? I would expect that there are countless permutations for ca. 20 muscles and some constraining criteria were likely applied.
It would be really helpful to outline the stochastical refinement approach in more detail and list the initial assumptions. I know that this could be gleaned from the referenced publications but having the information available will considerably increase the readability.

Secondly, it would be useful to have a figure showing the muscle attachments. I understand that this is provided in the first author's PhD thesis and is being prepared as a separate publication. However, it is impossible to get a good understanding of the insertions/origins from figures 1-5. I don't expect a full explanation and justification for every single muscle reconstructions as in the PhD thesis, but a figure with the mapped attachments (also in the thesis) is necessary here.

Thirdly, in the load case setup, explicit assumptions are made regarding the rotation of the humerus and femur along the long axis. The justification is provided in a referenced publication, but some brief explanation would be helpful here as well (in particular, considering that the referenced paper seems to be very niche and realistically inaccessible). Are other rotational settings possible or likely as well? What about rotation around the articulation on a horizontal plane?

Validity of the findings

The results are interesting and sound. In particular, the measured changes in muscle length are useful and a versatile approach to constrain hypothesised muscle reconstructions. Similar approaches have been used in other studies and mostly using digital methods, but the analogue approach offers a workable alternative solution.
The presentation of the results is sometimes a bit sparse.
- Figs. 2-5 are very busy and difficult to digest. A figure with some schematics illustrating the muscle vectors for the three tested scenarios would be very useful. I don't expect that all views are replicated from the photographs but the authors may consider a few simplified diagrams.

Figure 6: I assume the numbers refer to specific muscles but a key explaining this is missing.

Figure 7: Why are the FEA contour plots for the different scenarios superimposed? Space should not be a constraint with the journal and separate plots for upstroke and downstroke would be good to understand the differences in the stress regime for the different positions. The results for the muscle forces are based on the stress regime so having the separate plots would/should show the differences.

Additional comments

Overall, this is an interesting and useful study to provide new support for the often-discussed swimming style of plesiosaurs. The methodological approach is sound, but the readability is often hampered by the lack of detail. Underlying assumptions and explanations are often reduced to references to other (non-peer-reviewed) publications. Providing the respective information will make the manuscript fully self-contained and increase readability.
I have opted for major revisions to allow for sufficient time to implement the suggested changes.

·

Basic reporting

The use of English is uneven and improvable in the Results and Discussion parts of the manuscript. However, some unusual phrases are delightfully direct and clear, and I urge the authors to keep them. Some punctuation is correct but can be changed to English conventions (quotes, decimal points).

The manuscript's references and structure (both technical and conceptual) are excellent.

Experimental design

The study integrates two exciting approaches: full-size physical models and FE structural analysis. The physical articulation of the flippers with thread-based muscle paths is a superbly tangible and direct stage of muscle reconstruction. FESA-based optimization towards a compressive stress regime is spectacularly innovative for refining inferences of both anatomy and function, which is the major research question of the paper. The authors accounted for cartilage at proximal articulations.

Validity of the findings

The findings are rigorous and exciting. I suggest additional citation of the successes of FESA for functional morphology, including substantiated predictions of anatomy in living animals. I'm a huge fan of the method, and more evidence will further promote understanding in the community.

The conclusions are up to PeerJ standards.

Additional comments

The biggest improvements to the manuscript will be clarification of the figures. The photographic images are welcome, but the lines of action and labeling are cluttered, hard to see, or minimal. Currently these figures show the basic physical results, but the muscles will be more obvious if the threads are traced over with color-coded lines, with colors representing different muscles. The muscle reconstructions are the most substantial results in the paper, and Cryptoclidus deserves more visually bold and clear visualizations.

I would prefer more FESA figures, but understand that the emphasis is on muscle reconstructions.

Substantial suggestions and edits are on the commented manuscript. An often-repeated request is to spell out lines of action (LOA) more often, the first time it's used in a paragraph and in figure captions. Make the text as easy as possible for a reader to understand at a glance anywhere in the manuscript, without looking around for definitions of acronyms.

---

## Round 0.2 · Minor Revisions

Dear authors,

I would like to apologise for the long delay in this manuscript returning to you. None of the original three reviewers came back, and another I requested also never replied.

Overall I am very happy with how you responded to the reviewers' comments and suggestions. It has increased the readability of the text, the new figures are a big improvement and further explaining the methodology is better.

Given that, I only have wording and grammatical changes to suggest. Please note that PeerJ does not provide a full linguistic service during the proofing-stage so authors have the responsibility to ensure readability.

That being said, your manuscript is well-written. There are a few instances of grammatically 'clunky' sentences and phrases: "After this specialization of locomotion evolved" in the abstract for example. I'd like to give you a final chance to read through the text to make sure you're happy with it (especially given the number of track changes in the submitted version).

Nit-picking: "Oxford Clay" should be referred to as "Oxford Clay Formation".
From line 1099: "Pennate muscles can develop relatively high forces while their muscle lengths behave ", should that not be "generate" instead of develop?
From lines 1360, 2695, 4514: "crocodilian" should be "crocodylian".
From line 2090: "inhomogenities " should be "inhomogeneities".

References:
Krahl & Witzel 2021 - "Cryptoclidus eurymerus" isn't in italics.

---

## Round 0.3 · accepted · Accept

Dear authors,

Thank you for your revised manuscript. I am happy to accept it for publication in PeerJ.

In due course, the production team will contact you to take you through the proofing stages.

Thank you again for choosing PeerJ as your publication venue, and I hope you will choose us again in the future.